# A Global Black Carbon Dataset of Column Concentration and Microphysical Information Derived from MISR Multi-band Observations and Mie Scattering Simulations

Zhewen Liu[1], Jason Blake Cohen[1*], Pravash Tiwari[1], Luoyao Guan[1], Shuo Wang[1], Zhengqiang Li[2], Kai Qin[1]

[1]School of Environment Science and Spatial Informatics, China University of Mining and Technology, Xuzhou, 221116, China
[2]State Environmental Protection Key Laboratory of Satellite Remote Sensing, Aerospace Information Research Institute, Chinese Academy of Sciences, Beijing 100101, China

*Correspondence to*: Jason Blake Cohen (jasonbc@alum.mit.edu; jasonbc@cumt.edu.cn)

**Abstract.**

Black carbon, a major absorbing component of atmospheric aerosols, plays an important role in climate regulation, air quality, and human health, yet its column concentration and microphysical properties at regional and global scales remains highly uncertain. In this study, we implement an integrated approach that combines multi-angle, multi-band observations from the Multi-angle Imaging SpectroRadiometer (MISR) with a Mie scattering framework to estimate black carbon column properties including size and mixing state globally on a daily basis. By constraining particle size distributions with absorption aerosol optical depth and single scattering albedo across all four bands, the method simultaneously retrieves number and mass concentrations. Long-term simulations from 2005 to 2020 reveal distinct spatial and temporal patterns, with particularly high levels over biomass burning regions in Africa and South America as well as industrial and urban centers in Asia. Comparisons with ground-based sun photometer measurements and reanalysis data confirm the robustness and accuracy of the estimates. The resulting dataset provides a consistent global record of black carbon column concentrations, offering valuable support for constraining climate models, improving assessments of aerosol radiative forcing, and informing targeted mitigation strategies. The dataset is publicly available at https://doi.org/10.6084/m9.figshare.30173917 (Liu et al., 2026).

## 1 Introduction

Atmospheric aerosols, a mixture of fine solid and liquid particles widely distributed within the troposphere, play a pivotal role in climate systems, air quality, and human health due to their complex sources and heterogeneous distribution. Among these, black carbon (BC), characterized by its unique optical and physicochemical properties (Hodnebrog et al., 2014; Lund et al., 2018), has garnered increasing attention in atmospheric science (Jacobson, 2001; Li et al., 2016). BC's strong solar radiation absorption capacity makes it potentially among the most critical climate forcing agents after carbon dioxide. The

exact magnitude of its radiative impact remains a subject of ongoing debate in recent assessments (e.g., IPCC AR6) as well as many papers not cited in the AR6 (including but not limited to Mann et al. (2010), Chen et al. (2022), Everett et al. (2022), Kelesidis et al. (2022), Sedlacek et al. (2022), Ramachandran et al. (2023). Major causes of these differences are due to complexities in terms of both microphysical representations of BC on a per-particle basis, as well as emissions and column

loading of BC on an atmospheric basis, meaning that the discussion continues to have profound implications for global warming and regional climate change, as well as on the remote sensing retrieval community (Ramanathan and Carmichael, 2008; Li et al., 2022a). Predominantly generated through the incomplete combustion of carbonaceous materials (Bond et al., 2013; Xie et al., 2025), BC sources include natural events (Cohen, 2014), such as volcanic eruptions and wildfires, and anthropogenic activities (Szidat et al., 2006; Lin et al., 2020b; Wang et al., 2020), such as fossil fuel combustion, industrial

sources, steel and other materials production, and vehicular emissions. Freshly emitted BC particles are typically independent entities (Tanaka et al., 2012), but aging processes in the atmosphere lead to interactions with other primary aerosols, secondary aerosol precursors, and atmospheric water vapor, forming a "core-shell" type of structure that is thermodynamically stable (Chen et al., 2013; Wang et al., 2021b). This aged aerosol is the dominant type found in the in-situ environment (Roberts and Jones, 2004; Mylläri et al., 2019) especially so in more heavily polluted areas (He et al., 2015).

Such a configuration amplifies BC's light absorption capabilities (Hansen et al., 2007; Chung et al., 2012; Cappa et al., 2012; Tiwari et al., 2023), further intensifying its impact on radiative balance and atmospheric dynamics, especially regionally (Lund et al., 2017; Wang et al., 2019).

Additionally, BC deposition on snow and ice surfaces reduces albedo (Zhao and Garrett, 2015; Zhao et al., 2023), accelerating melt rates and exerting a significant influence on the climate(Xie et al., 2018), particularly in high-latitude and

high-altitude regions. Beyond its climatic impacts (Rosenfeld et al., 2014; Ding et al., 2016; Ma et al., 2020), BC disrupts boundary layer structures, facilitates haze formation, and inhibits pollutant dispersion, thereby degrading air quality (Guo et al., 2019). Furthermore, BC particles carry toxic combustion byproducts, posing relatively more severe health risks than $PM_{2.5}$ on average, including increased respiratory and cardiovascular disease incidences (Cassee et al., 2013; Yang et al., 2021), representing a significant public health concern. Therefore, enhancing BC monitoring and research is essential for

quantifying its sources, evolution, and impacts, improving climate model predictions, and formulating targeted pollution mitigation and emission reduction strategies.

The column concentration of BC, an essential indicator of its total amount in the atmospheric column, has been widely used to evaluate BC's radiative effects and climate impact. Advances in ground-based observation, satellite remote sensing, and numerical simulations have significantly enriched the understanding of its spatiotemporal distribution and environmental

consequences (Liu et al., 2024a, b; Tiwari et al., 2025). Ground-based measurements using instruments such as sun photometers, BC analyzers, and aerosol optical property devices provide high temporal resolution and measurement accuracy but are spatially confined to specific regions (Schwarz et al., 2008; Lee et al., 2016b). Seasonal variations in BC column concentrations are evident (Li et al., 2023; Yu et al., 2024), with studies in high-pollution areas indicating substantially higher levels during winter compared to summer, or during intense periods of industrial activity and/or local

biomass burning. However, the limited spatial coverage of ground observations fails to comprehensively represent global BC column concentration distributions.

Satellite remote sensing has opened new avenues for monitoring BC column concentrations at regional and global scales (Yu et al., 2024). Traditional approaches have frequently used retrievals of aerosol optical depth from satellite observations to derive BC column mass concentrations using fixed and/or simplified microphysical properties of BC (Junghenn Noyes et al.,

2020; Li et al., 2020, 2022b; Limbacher et al., 2022; Torres et al., 2020; Zhang et al., 2025). While sophisticated component-based retrieval frameworks (Li et al., 2019) have made advancements in this field including better use of polarization and improved representation of particle size, the use of multi-angle and multi-band observations from MISR provide a unique possibility to further constrain the microphysical variability of BC, such as high resolution particle size and mixing state variability, and non-homogenous computation of SSA, allowing both more realistic per-particle microphysical and total

atmospheric column constraints on BC on a global scale. However, challenges in spatial resolution and data accuracy persist, especially in regions with complex terrain, severe pollution, where the sources of BC emissions are undergoing rapid changes, where different sources of pollutants mix with each other, or in other locations where retrieval reliability and uncertainty requires further improvement (Wang et al., 2021c; Lu et al., 2025). Numerical simulations have been proven invaluable in investigating the sources, transport, and deposition processes of BC column concentrations (Randles et al.,

2017; Bousserez et al., 2020; Wang et al., 2021a). Atmospheric chemical transport models and climate models reveal that long-range BC transport contributes significantly to high concentrations in certain regions and exerts substantial downstream climatic and environmental impacts (Wang et al., 2017; Tegtmeier et al., 2022; Yang et al., 2022; Senf et al., 2023; Zhao et al., 2024; Chakraborty et al., 2025). Despite their spatial and temporal comprehensiveness, the accuracy of numerical simulations heavily depends on emission inventories in both space and time (Wang et al., 2025; Li et al., 2025), in addition

to the understanding and parameterization of, as well as access to sufficient data to compute relevant physicochemical processes (Chen and Prinn, 2006; Kim et al., 2008).

This study proposes an innovative methodology for estimating BC column concentrations by integrating satellite remote sensing data, aiming to enhance calculation efficiency and accuracy through the effective utilization of multisource satellite datasets. Specifically, the method involves leveraging data from four spectral bands of the MISR satellite by reading and

processing them on a daily basis to obtain critical parameters (Diner et al., 1998; Ahn et al., 2008; Lee et al., 2016a), including Absorption Aerosol Optical Depth (AAOD) and Single Scattering Albedo (SSA). These parameters are then input into the Mie scattering model (MIE model) for particulate size retrieval, followed by BC column concentration estimation. This estimation encompasses both mass and particle count data, providing a comprehensive basis for evaluating BC's environmental and climatic effects. Detailed sensitivity analyses of the data were conducted to ensure result reliability and

stability. Furthermore, comparisons with AERONET optical properties and inter-comparisons with the MERRA-2 reanalysis serve to evaluate and demonstrate the method's consistency, realism over areas with limited, misplaced, or out of date a priori emissions data, and applicability to offering a new consistent product across global scales at daily resolution over decadal temporal scales. This integrated approach, combining satellite remote sensing and model computations, presents an

innovative solution for BC column concentration estimation, improving data processing efficiency and mitigating the limitations of traditional single-source methods. By effectively leveraging multi-band satellite observations, this methodology holds promise for advancing aerosol monitoring and climate model validation, offering robust technical support for regional and global BC research.

## 2 Materials and methods

### 2.1 MISR AOD/SSA/AAOD

The MISR, a key component of NASA's Earth Observing System (EOS), captures multi-angle reflected sunlight images to investigate Earth's ecosystems and climate change (Kalashnikova and Kahn, 2006). Launched aboard the Terra satellite in 1999 (Shi and Cressie, 2007), MISR consists of nine push-broom cameras that image Earth in four spectral bands (443, 555, 670, and 865nm) and provide global coverage every nine days (Martonchik et al., 2004). Multi-angle observations mitigate the influence of vertical aerosol distribution heterogeneity on aerosol optical depth (AOD) retrievals and aid in distinguishing optical properties of various aerosol types based on their sphericity (Wang and Gordon, 1994; Kalashnikova and Kahn, 2008; Kalashnikova et al., 2013).

The product used in this study, MIL3DAEN_4, is MISR's Level 3 global aerosol product (CGAS), providing aerosol information on a global 0.5° × 0.5° latitude-longitude grid (NASA/LARC/SD/ASDC, 2008). These data are aggregated from higher-resolution (4.4 km × 4.4 km) MISR Level 2 aerosol retrievals and include daily, monthly, seasonal, and annual averages (Garay et al., 2017; Si et al., 2020). For each temporal product, the grid-cell value is computed by aggregating all screened Level-2 retrieval samples falling within the corresponding 0.5° × 0.5° cell during that reporting period. A key feature of the MIL3DAEN_4 product is its capacity for global-scale aerosol monitoring and multi-temporal data aggregation. By utilizing multi-angle imaging, it provides AOD and AAOD data, effectively capturing both scattering and absorbing properties of aerosols. In MIL3DAEN_4, the gridded data resolution is 0.5° × 0.5°, with each grid cell value averaged from higher-resolution Level 2 product samples, assigning equal weight without regard to sampling frequency. Specifically, all screened samples within a grid cell are used to compute the arithmetic mean, with equal weight assigned to each valid sample. To maintain data credibility, the MIL3DAEN_4 product only includes samples where the "Aerosol_Retrieval_Screening_Flags" are set to 0, thereby excluding samples failing the standard screening criteria and typically reducing coverage over persistently challenging surfaces (e.g., Greenland and Antarctica). The MIL3DAEN_4 product inherits a "Stage 3 Validated" maturity level, with reported uncertainties of approximately ±0.05 (or 20% × AERONET) for AOD (Kahn et al., 2010; Garay et al., 2020), based on extensive global evaluations.

Moreover, MIL3DAEN_4 offers substantial temporal and spatial coverage, supporting analyses of aerosol temporal trends and spatial distributions. Accumulating long-term data, MIL3DAEN_4 facilitates examination of aerosols' long-term trends and their potential impacts on climate and ecosystems. Despite its relatively coarse spatial resolution, MIL3DAEN_4 provides unique insights into aerosols' role within the Earth's climate system on both global and regional scales. Compared

with other aerosol remote sensing products, MISR's multi-angle approach enables improved retrieval accuracy of aerosol optical properties, particularly for specific regions and aerosol types.

In summary, the MIL3DAEN_4 product, as MISR's Level 3 aerosol product, aggregates aerosol optical and physical characteristics in a globally gridded format, offering multi-temporal data on aerosol variations with significant scientific value. MIL3DAEN_4 plays a critical role in studies of aerosol scattering and absorption characteristics. Although limited by its spatial resolution, its multi-angle and multi-spectral observational capability provides robust data for monitoring aerosols at varying spatial and temporal scales.

## 2.2 MIE model

The Mie model is a theoretical model used to calculate the interaction between electromagnetic waves and spherical particles and holds significant applications in aerosol optical property research (Bohren and Huffman, 1998). Named after the German physicist Gustav Mie (1908), the Mie scattering model is based on Maxwell's equations and addresses the scattering and absorption of incident electromagnetic waves by spherical particles to determine properties such as the intensity, phase, and polarization of scattered light. This model considers factors such as the wavelength of incident radiation, particle size (expressed through the size parameter $x$, where $x = \pi D / \lambda$, with D as particle diameter and $\lambda$ as wavelength), and the particle's complex refractive index ($m = n + ik$, where $n$ is the real part and $k$ is the imaginary part of the refractive index).

Using a "core-shell" aerosol structure, the Mie scattering optical model is applied to calculate radiative parameters for extinction aerosols (Kahnert et al., 2007; Wang et al., 2021b; Liu et al., 2024b). This model, based on Mie scattering theory, treats light waves as electron waves and solves Maxwell's equations for both internal and external regions of the particle under boundary conditions determined by particle shape and size. Mie scattering theory thus provides an exact solution for a homogeneous, spherical particle in the far-field under monochromatic plane wave illumination. Based on the particle's chemical composition, size parameter, and refractive index, the model calculates the scattering, extinction, and absorption efficiencies ($Q_{sca}$, $Q_{ext}$, $Q_{abs}$), which are then used to determine the optical properties of BC particles across MISR's four spectral bands. This methodological approach focuses on these core physical outputs and their application within the retrieval framework, while the detailed mathematical derivations using Bessel functions are extensively documented in established literature(Abramowitz and Stegun, 1965; Ladutenko et al., 2017).

## 2.3 Uncertainty Characterization of Comparison Datasets

To enhance the completeness of the analysis and provide a broader contextual framework, this study incorporates comparisons with ground-based AERONET observations and MERRA-2 reanalysis data. These reference datasets are utilized to situating the retrieval results within the context of established aerosol products rather than to serve as an absolute evaluation of accuracy. It is essential to acknowledge that both reference datasets possess inherent uncertainties and physical limitations that vary spatiotemporally, which must be considered when interpreting discrepancies.

The MERRA-2 monthly product provides a global spatial context for validation but should be interpreted as a model-driven estimate constrained by observations rather than direct observation. MERRA-2 assimilates total column AOD from satellites (MODIS, MISR) and ground stations (AERONET) but does not assimilate BC concentrations directly (Buchard et al., 2017; Randles et al., 2017). The apportionment of the assimilated total AOD into specific aerosol components (BC, dust, sulfate, etc.) is governed by the GOCART aerosol module's physics and the underlying emission inventories. Previous validation studies have indicated that MERRA-2 tends to underestimate BC surface concentrations in regions of high anthropogenic emissions, such as East Asia and South Asia, often by 20–50% due to uncertainties in emission inventories (Song et al., 2018; Qin et al., 2019; Yan et al., 2022). Conversely, over remote oceans and in the stratosphere, MERRA-2 has been shown to potentially overestimate BC loading compared to aircraft observations (Buchard et al., 2017), while studies considering active aging of BC tend to not demonstrate a similar overestimation in these regions (Cohen et al., 2011).

The AERONET Version 3 (V3) Level 2.0 inversion products serve as the primary ground-based standard for validating aerosol optical properties(Sinyuk et al., 2020). While AOD is measured with high accuracy (with an uncertainty of +0.06 to - 0.02) (Eck et al., 2001; Giles et al., 2019), the retrieval of absorption properties such as SSA and AAOD is subject to greater uncertainty, particularly under low aerosol loading conditions. Sinyuk et al. (2020) demonstrated that the uncertainty in AERONET SSA retrievals is approximately $\pm0.03$ when $AOD_{440nm} > 0.4$. However, as aerosol loading decreases below this threshold, the sensitivity of sky radiance measurements to absorption diminishes, causing SSA uncertainties to increase to $\pm0.05$ or higher when $AOD_{440nm} < 0.2$ (Dubovik et al., 2000; Schutgens et al., 2021; Zhang et al., 2025). Consequently, discrepancies between MISR retrievals and AERONET data in clean background regions (low AOD) may largely reflect the noise floor of the ground-based inversions rather than satellite retrieval errors. To align with these physical sensitivity constraints and mitigate the impact of potential noise floor issues in ground-based inversions, this study exclusively utilizes MISR data where the AOD at 443nm is greater than 0.25. This threshold ensures that the absorption signals analyzed are physically robust and that the comparisons remain scientifically meaningful even in regions with varying aerosol concentrations, at the cost of a reduction in overall spatial coverage.

Additionally, the inherent differences between MISR (which provides clear-sky, instantaneous overpasses) and MERRA-2 (which provides all-sky, diurnal averages) introduce temporal and spatial representativeness errors. Therefore, this inter-comparison is primarily intended to provide more possibilities for understanding the global spatiotemporal variability of BC. By integrating results from different observational and modeling platforms, this approach offers a more comprehensive perspective on the reliability and applicability of the proposed framework across diverse atmospheric environments.

**2.4 Data Selection Strategy and Regional Partitioning**

To provide comprehensive coverage over a long timeframe and minimize uncertainties associated with the integration of multiple data sources, this study utilizes multispectral remote sensing retrievals from MISR. Compared to other remote sensing sources, MISR offers an extensive observational record since 2000 and provides data across four spectral bands, which serves as an important foundation for accurately calculating BC-related parameters. The accuracy of MISR products

has been established through extensive global validation against AERONET, providing a physical foundation for our retrieval. In terms of specific error budgets, AERONET reports uncertainties of 0.03 for SSA and +0.06 to -0.02 for AOD. In comparison, the MISR AOD uncertainty is established at +0.05 or $0.2 \times$ AERONET AOD. Given these constraints, the MISR SSA uncertainty is estimated to be at least as large as the AERONET SSA uncertainty, translating to approximately 20%–40% under typical conditions where this process is successful, the SSA ranges from 0.80 to 0.95. To further ensure the robustness of our inputs, we performed a consistency check by comparing MISR-derived AAOD with AERONET observations (Figure B1 and Table C4). The resulting differences have Root Mean Square Error (RMSE) and Mean Absolute Error (MAE) statistics both falling within approximately ~64% (assuming an SSA of 0.92), which is the combined uncertainty range (propagated using 20% AOD and 20%–40% SSA uncertainties). This alignment with ground-based benchmarks confirms that the input parameters provide a solid basis for reliable BC estimation. The multi-angle observation capability of MISR is highly effective for identifying aerosol optical properties, particularly in retrieving key parameters such as AAOD, where the multi-angle constraints significantly enhance estimation accuracy.

The criterion that $AOD_{443nm} > 0.25$ reflects findings from previous studies showing that when AOD is relatively high, AAOD measurements become more reliable, thereby reducing observational uncertainty (Mallet et al., 2017; Schutgens et al., 2021; Zhang et al., 2025). Consequently, only the subset of MISR pixels from October 2004 to December 2020 meeting this criterion were selected. This filtering method enables the acquisition of more reliable BC data for analyzing its global distribution and spatiotemporal variation trends. For consistency in annual statistics and interannual trend analysis, the year-by-year evaluation is restricted to complete calendar years (2005–2020). This approach ensures comparability across years and avoids biases associated with partial-year data in 2004, although all individual days of data are retained in the complete dataset.

To improve analytical efficiency and account for regional aerosol characteristics, this study adopted a global regional partitioning method based on Cohen and Wang (2014). As shown in Figure 1, the globe is divided into nine major regions (North America, South America, Europe, Africa, East Asia, Southeast Asia, South Asia, North Asia, Oceania), addressing both the global-scale requirements and the need to adapt to the specific climate and geographic features. Additionally, adjustments were made to the regional boundaries for parts of Australia, Southeast Asia, and East Asia to better reflect the unique climate conditions and aerosol emission characteristics of these areas, ensuring more precise estimation of BC parameters in these critical regions.

This systematic regional division strategy facilitates the optimal use of MISR's long-term observational record to analyze spatiotemporal distribution patterns of BC across diverse regimes, providing a solid scientific basis for further climate and environmental analysis.

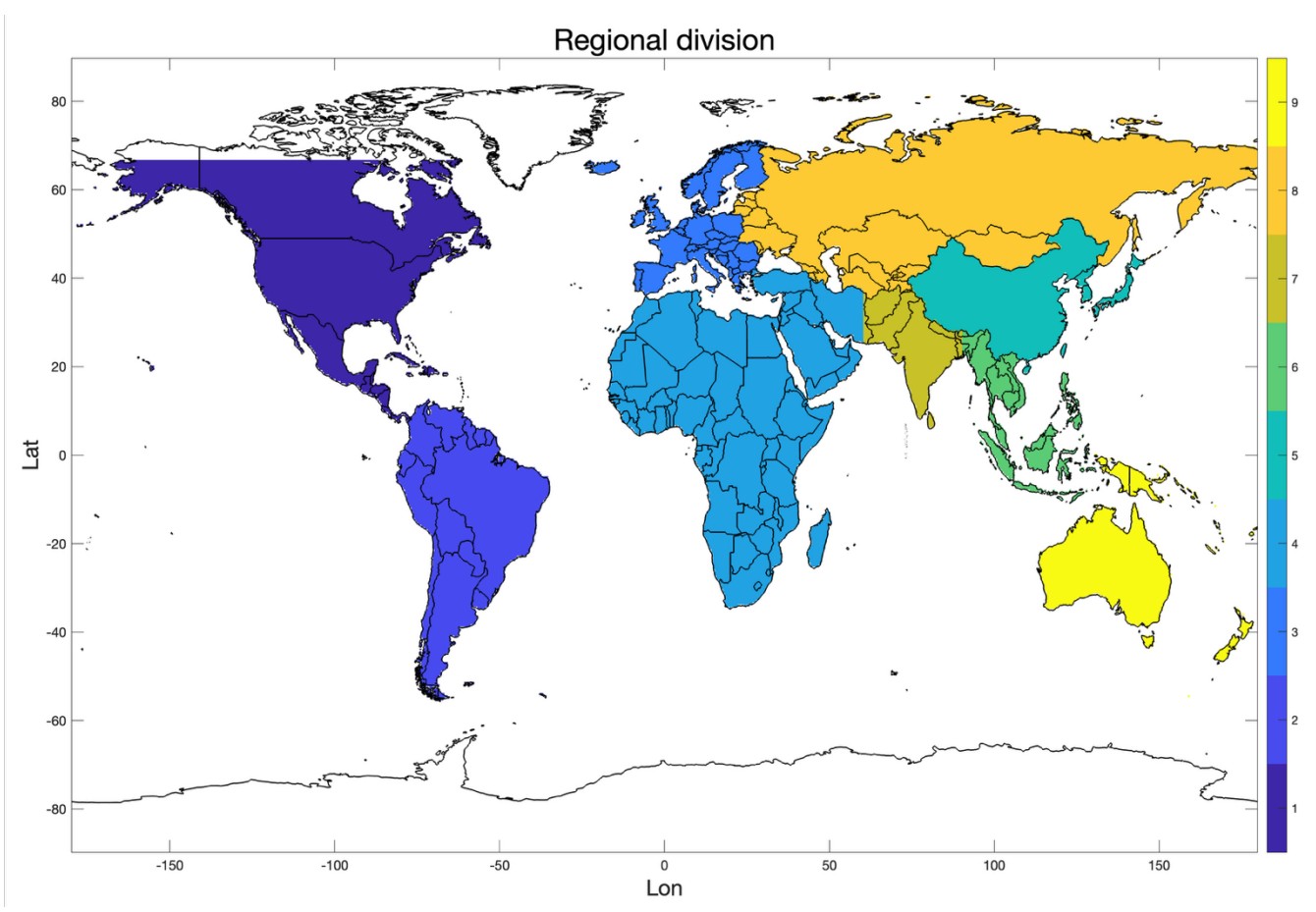

**Figure 1. The nine geographic spatial defined regions used in this study.**

## 2.5 Black Carbon Column Concentration Retrieval Framework

To accurately simulate the influence of aerosol microphysical structures on optical properties, this study implements an integrated retrieval framework based on a "core-shell" Mie model. This configuration accounts for the realistic atmospheric state where the absorptive BC core is frequently encapsulated by non-absorbing or scattering materials through aging and mixing processes. Within this framework, absorptive aerosols are assumed to consist of a BC core and a scattering coating (sulfate shell). The complex refractive index (CRI) of the BC core is prescribed as $2.0\pm1.0i$ (Schuster et al., 2005), and the CRI of the shell is assigned as $1.52\pm5\times10^{-4}i$ (Aouizerats et al., 2010). The simulation setup selects a "core" particle size range of 50–500 nm (in 10 nm increments) and a "shell" size range of 50–1000 nm (also in 10 nm increments). Utilizing this size range combination, a comprehensive database of simulated optical properties (including $Q_{sca}$, $Q_{ext}$, $Q_{abs}$) was generated across MISR's four spectral bands.

The SSA was used as the primary observational constraint to limit the range of microphysical solutions at each spatial grid point on a daily basis. SSA values derived from MISR's four bands were incorporated as wavelength-dependent boundary

conditions in the simulations, enabling validation of the simulated optical properties and providing objective criteria for selecting the final, physically consistent solutions. Observed values from each band defined a possible solution space by considering the ±0.03 uncertainty of SSA observations (Torres et al., 2020; Tiwari et al., 2025), to account for inherent observational uncertainties and instrument noise. This uncertainty level represents a conservative physical limit aligned with the AERONET ground-based benchmark of ~0.03, although we acknowledge that in fact it may be slightly larger. Under this framework, only grid points that met the range requirements across all four bands in tandem are retained. This multi-band constraint analysis method enhances the reliability of the simulation results and avoids biases that may arise from single-band analyses. Through this filtering process, we obtained a set of particle radii that meet the observation criteria, representing the plausible particle size ranges under specific conditions and laying a foundation for further analysis of aerosol scattering and absorption properties. This filtering process identifies the set of plausible particle radii ($R$) that are physically consistent with the observed atmospheric absorption signatures.

After the SSA-based screening, the remaining solutions define a physically plausible set of $R$ for each grid cell and day. For these retained solutions, Mie calculations simultaneously provide $Q_{abs}$ across various particle size combinations. Using these calculated $Q_{abs}$ and applying Eq.(1), we further derive the overall absorption coefficient ($\sigma_{abs}$) for individual particles. This quantity describes the capacity of an individual particle to absorb incident radiation at a given wavelength and serves as the direct link between microphysical properties and column-integrated absorption.

$$\sigma_{abs} = \left[\frac{\pi (2R)^2}{4}\right] \cdot Q_{abs} \qquad (1)$$

With $\sigma_{abs}$ determined, the MISR AAOD observations are then used to estimate the column number concentration ($N_{col}$) by inverting the relationship between column absorption and single-particle absorption, as expressed in Eq.(2). $N_{col}$ represents the total number of particles within the atmospheric column and is a key metric for characterizing aerosol loading and its spatiotemporal variability.

$$N_{col} = \frac{AAOD}{\sigma_{abs}} \qquad (2)$$

Furthermore, the particle column mass concentration ($M_{col}$) is derived from $N_{col}$ by applying Eqs.(3) and (4). Here, $\rho$ denotes the particle density (prescribed as 1.8 g cm$^{-3}$ (Bond and Bergstrom, 2006), and $M$ represents the mass of a single BC particle. This procedure establishes the quantitative basis for transforming AAOD-constrained optical retrievals into physically interpretable BC mass loading.

$$M = \rho \cdot \frac{4}{3}\pi R^3 \qquad (3)$$

$$M_{col} = N_{col} \cdot M \qquad (4)$$

Collectively, this workflow integrates the SSA-constrained microphysical solution space with particle absorption properties ($Q_{abs}$ and $\sigma_{abs}$), ultimately yielding column number and mass concentrations ($N_{col}$ and $M_{col}$). By synthesizing particle size,

absorption efficiency, and absorption cross-section into the resulting number/mass loadings, the framework provides a systematic and physically consistent pathway for retrieving the spatiotemporal distribution of BC and related absorptive aerosols, thereby supporting subsequent analyses of aerosol radiative effects and climate impacts.

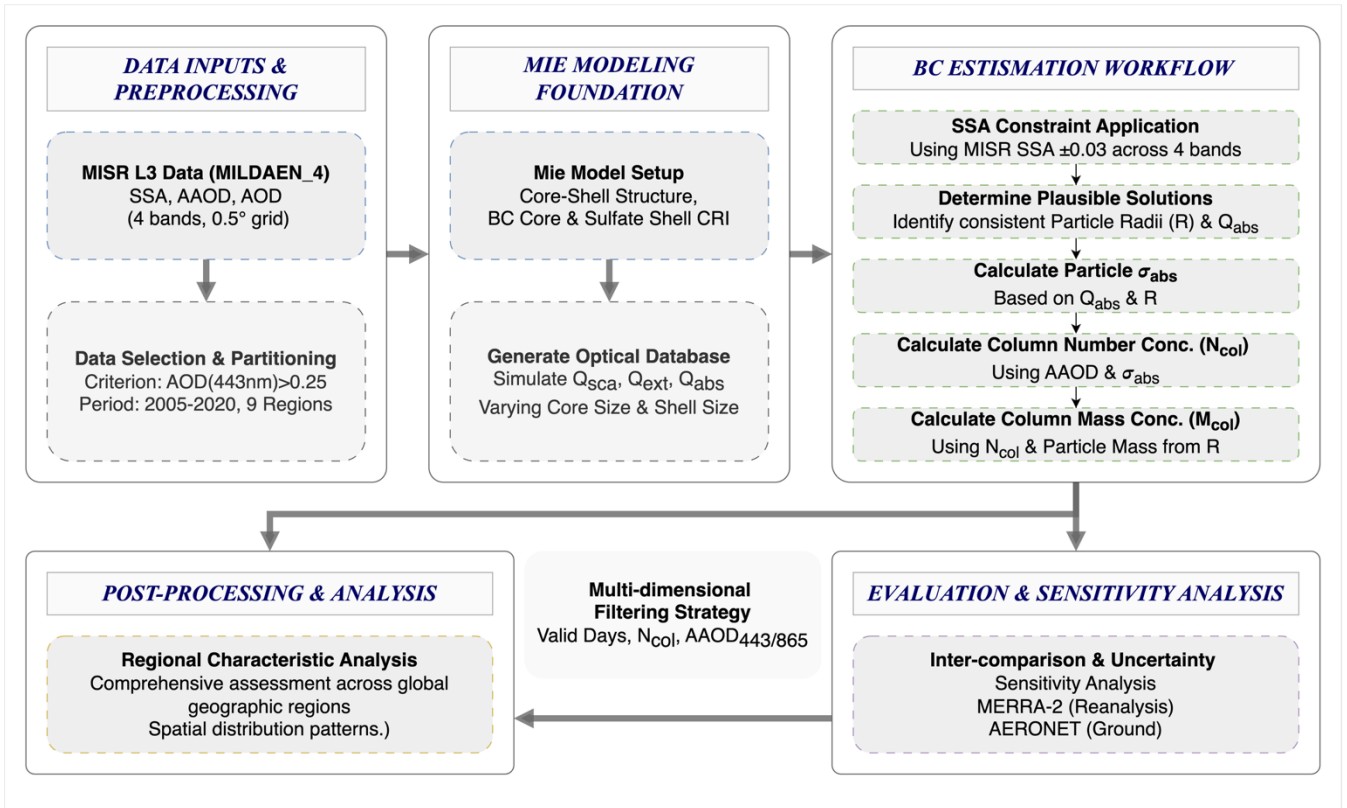

**Figure 2. Flowchart of the estimation of the global black carbon (BC) column concentration product based on MISR observations.**

### 2.6 Sensitivity Analysis and Inter-comparison Scheme

To evaluate the robustness of the retrieval framework across the prescribed Mie model parameter space, a statistical sensitivity analysis is performed for the 2005–2020 period. The procedure quantifies the variability of the retrieved BC column mass concentration (MASS), column number concentration (NUM), core particle size (BCsize), and shell size (Shellsize) by calculating the mean and specific percentiles (25th, 50th, and 75th) for each grid point. This statistical framework enables a systematic assessment of the retrieval stability across varying concentration regimes and particle size combinations, using probability density functions (PDFs) and time-series analysis to characterize the response of the retrieval to input parameter variations.

To facilitate a comprehensive evaluation of BC column mass concentrations derived from different data sources, this study implemented a rigorous matching and screening procedure between the MISR-derived products and MERRA-2 reanalysis data. Specifically, daily grid-level MISR retrievals first underwent a quality-filtering process to serve as the reference

standard, and the corresponding MERRA-2 data were then spatially and temporally co-located within the same $0.5° \times 0.5°$ geographic domain. This step ensures strict consistency in both spatial resolution and temporal scale across the two datasets. Based on this integrated matching framework, comparative analyses were stratified under different concentration regimes (e.g., low versus high levels) and particle size assumptions, ranging from small to large sizes. Such a daily grid-level matching strategy effectively minimizes systematic biases arising from differences in spatial and temporal aggregation methods, thereby ensuring that the comparisons between MISR and MERRA-2 are statistically representative and physically meaningful. Moreover, this approach provides a robust basis for systematically assessing the applicability and limitations of the two datasets across diverse regions, emission conditions, and climatic contexts.

## 2.7 Filtering Strategy for Representative Regional Analysis

Following the retrieval of BC column number and mass concentration products, a rigorous mass control and multi-dimensional filtering procedure was implemented to ensure the scientific robustness and reliability of subsequent analyses. This screening scheme is designed to identify representative spatial domains by introducing three primary physical constraints: the number of valid observational days, the BC column number concentration, and the $AAOD_{443/865}$ ratio.

First, to maintain temporal stability, a minimum threshold was imposed on the number of valid observational days. This constraint ensures that regional-scale statistics are not biased by occasional or transient events, thereby enhancing the representativeness of the dataset (Chen and Prinn, 2006). Second, BC column number concentration serves as a direct indicator of atmospheric loading; however, its distribution may contain outliers from extreme or abnormal events. To prevent these lower-confidence results from distorting spatial averages or obscuring underlying atmospheric signals, appropriate thresholding was applied to exclude clearly unrealistic pollution cases.

Second, the BC column number concentration is utilized as a key criterion to delineate and focus on typical pollution hotspots. Rather than merely removing outliers, appropriate thresholding—such as percentile-based criteria—is applied to extract regions with significant and sustained BC loadings. This process ensures that the subsequent analysis focuses on geographically continuous areas that represent major atmospheric pollution regimes, thereby excluding low-concentration zones where the BC signal is too weak to provide regional explanatory power. By selecting these high-loading typical regions, the framework ensures that the derived spatiotemporal characteristics are both physically meaningful and representative of large-scale emission sources.

Finally, the $AAOD_{443/865}$ ratio was incorporated as a spectral optical constraint to improve the specificity of the BC products (Bergstrom et al., 2007; Russell et al., 2010; Helin et al., 2021; Bigi et al., 2023). This ratio characterizes aerosol absorption across different wavebands and is particularly effective at identifying the presence of large, strongly absorbing non-BC aerosols, such as mineral dust. By filtering out grid points with abnormally high ratios, potential dust contamination was significantly reduced, ensuring that the retained data predominantly reflects the contribution of carbonaceous aerosols. Through the integration of these constraints, the reliability of the dataset was enhanced across temporal, concentration, and optical dimensions.

It is important to emphasize that regional differences in climatic backgrounds and emission structures necessitate the use of adaptive filtering strategies rather than a uniform global threshold. A uniform threshold across all regions may lead to excessive sample loss in some areas or insufficient removal of anomalous cases in others, potentially biasing the results. To address this, a region-specific approach was adopted, selecting appropriate percentile-based thresholds or quantile criteria according to local data distributions. This method not only maximizes data representativeness but also removes samples with limited explanatory power. By incorporating such regionally adaptive criteria, the dataset better reflects the true atmospheric processes and BC pollution characteristics across different environments. The complete methodological framework, including the retrieval process and the hierarchical logic of this filtering strategy, is illustrated in Figure 2. After these mass control and filtering steps, the final sample set is both stable and credible, providing a solid basis for in-depth investigations into BC concentration levels, spatiotemporal.

## 3 Results

### 3.1 Simulation of Aerosol Optical Properties Using SSA Constraints and Particle Size Combinations

Figures 3a and 3b depict the global distribution of BC core diameters and their standard deviations during 2004–2020. Larger BC particle sizes (indicated in red, ranging from 155 to 165 nm) are observed in central and southern Africa, parts of South America, forested and densely populated regions of North America, core urban areas of Europe, and much of Asia. In central and southern Africa, the standard deviation of BC sizes remains relatively balanced (orange, between 30 and 35 nm). The relatively large particle size in these regions is likely linked to widespread biomass burning (e.g., agricultural residue burning and forest fires), where BC generation tends to be stable, less affected by human disturbances, and associated with relatively uniform pollution sources. By contrast, in Europe and large parts of Asia, the standard deviation of BC size is considerably higher (orange to red, 40–45 nm). In Europe, this is primarily attributed to industrial activities and traffic emissions related to high population density, whereas in Asia, in addition to industrial and demographic factors, biomass burning from agriculture, domestic activities, and wildfires also plays an important role. The pronounced variability in particle size in these regions reflects the destabilizing effects of anthropogenic production processes and the uncertainty of wildfire occurrence on BC formation. In North America, BC size variation is more irregular, highlighting the complexity of biomass burning emissions caused by sporadic forest fires, while urban areas exhibit patterns comparable to those in Europe. In South America, regions with larger BC particle sizes also display greater variability: urban centers correspond to anthropogenic sources, whereas in and around the Amazon rainforest, larger BC particles are sporadically observed, likely resulting from frequent deforestation and burning, with biomass burning showing a non-cyclical nature.

The shell size of particles reflects the extent to which BC cores accumulate secondary substances in the atmosphere. As shown in Figures 3c and 3d, relatively large shell sizes (red, exceeding 710 nm) are concentrated in southeastern Africa, northern South America, and across South, Southeast, and East Asia. The increase in shell size in these regions is likely

350 associated with the mixing of BC with other aerosol species, particularly during biomass burning and agricultural waste processing. For instance, in northern Amazonia, shell sizes are substantial but exhibit little variability, suggesting persistent biomass burning as a long-term source with relatively low emission fluctuations. Similar patterns are also found in the central African grassland regions. In parts of Asia, such as China, India, and Southeast Asia, shell sizes generally exceed 690 nm. In Southeast Asia, although the distribution is patchy, the variability remains relatively modest. This can be attributed to

355 frequent, stable biomass burning and intensive emissions from mining and industrial activities, which facilitate the adsorption of additional pollutants, leading to larger shells. The scattered higher variability in Southeast Asia may be linked to seasonal shifts in emissions, particularly differences in biomass burning activities between wet and dry seasons. By contrast, in northern Europe and urban areas of North America, shell sizes are relatively smaller (blue, below ~650 nm) and exhibit low standard deviations (blue to green), indicating greater stability. This pattern likely reflects the effects of stricter

360 environmental regulations and emission controls, which suppress shell growth, reduce the particles' capacity to adsorb additional material, and minimize fluctuations in pollutant emissions.

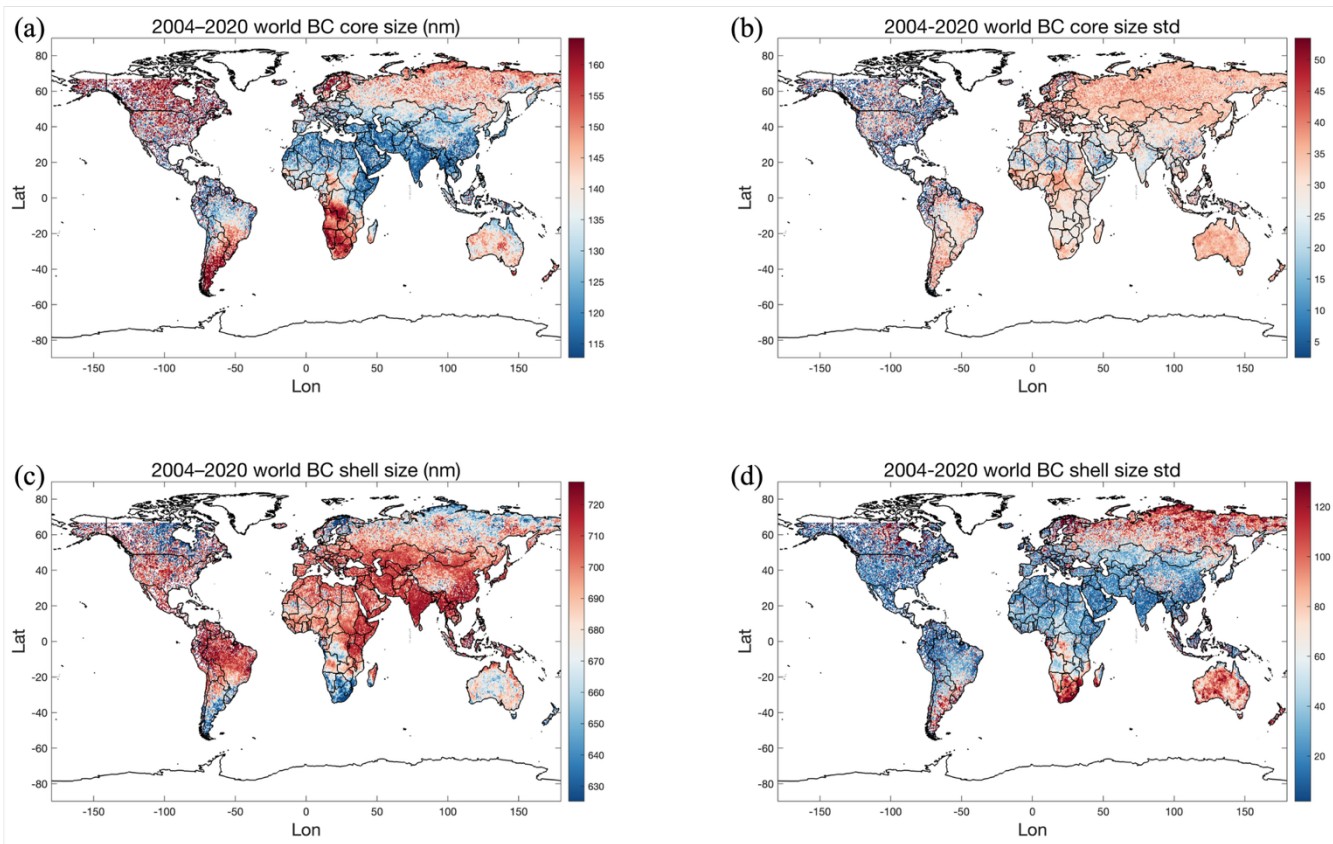

**Figure 3. Schematic diagram of global terrestrial products with BC core and shell particle sizes from 2004 to 2020.** (a) Mean size of BC core. (b) BC core size standard deviation. (c) Mean size of BC shell. (d) Standard deviation of BC shell size.

## 3.2 Global Analysis of Black Carbon Column Mass and Number Concentrations

Figures 4a and 4b illustrate the global distribution of BC column mass concentration and its standard deviation from 2004 to 2020. Pronounced BC mass concentrations, exceeding 8000 kg per grid per day, are observed over central and southern Africa, parts of North America, and central to southern South America. These elevated concentrations are primarily attributed to biomass burning. In contrast, regions such as the Amazon rainforest exhibit moderate BC concentrations, typically ranging between 6000 and 8000 kg per grid per day, which may be linked to agricultural activities and deforestation. Several densely populated areas of Asia, including China, Japan, and Southeast Asia, also show relatively high BC mass concentrations, reflecting the combined influence of industrial activity, transportation emissions, and agricultural practices. Conversely, most regions in Europe and Australia report comparatively low BC concentrations, generally below 6000 kg per grid per day, a pattern likely shaped by stringent air pollution controls that effectively limit BC emissions. Central and western Africa also stand out with both high emission intensity and substantial standard deviations (around 8000 kg per grid per day), indicating strong interannual variability associated with seasonal differences in biomass burning. Elevated variability is likewise observed in parts of East and Southeast Asia, as well as northern Australia, suggesting the combined influence of multiple pollution sources and fluctuating emission dynamics. By comparison, much of South America exhibits moderate standard deviations, pointing to relatively greater temporal variability, while regions such as southern Australia and much of Europe display lower variability, consistent with more stable emission conditions.

Figures 4c and 4d present the global distribution of BC column number concentration and its standard deviation during the same period. Overall, the spatial distribution of number concentration broadly mirrors that of mass concentration, with notable differences in northwestern Africa, the Arabian Peninsula, and South Asia, where number concentrations appear relatively higher. This discrepancy likely reflects spatial heterogeneity in emission sources as well as differences in BC particle size distributions. In the northwestern African region, high population density and industrial activity are major contributors to the emission of smaller BC particles. In South America, areas characterized by high mass concentrations also exhibit similarly extensive coverage in number concentration, underscoring the role of biomass burning and agricultural activities in generating and dispersing large quantities of BC particles. Regarding variability, central and northwestern Africa exhibit pronounced fluctuations, indicating marked temporal inconsistency in emissions. In South America, variability in number concentration is largely confined to limited areas such as Brazil, São Paulo, and scattered southern regions, while the overall continent demonstrates relatively stable BC emissions.

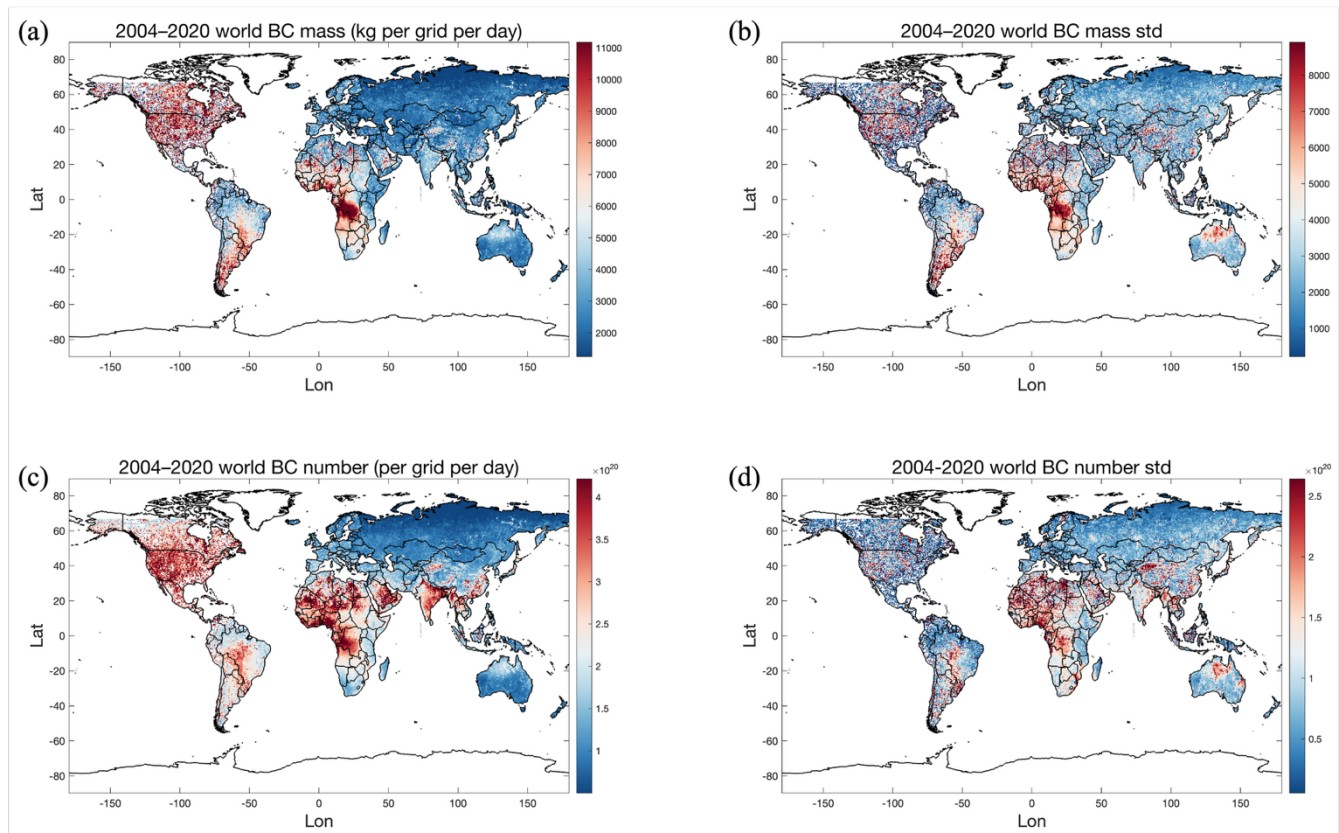

**Figure 4. Schematic diagram of global terrestrial products with BC column mass and number concentration from 2004 to 2020.** (a) Mean mass concentration of BC column. (b) Standard deviation of BC column mass concentration. (c) Mean number concentration of BC columns. (d) Standard deviation of BC column number concentration.

As shown in Table 1, the annual mean BC column mass concentrations are reported for nine regions, while the corresponding column number concentrations, core sizes, and shell sizes are provided in Tables C1, C2, C3. For North America, BC column mass concentrations increased during 2005–2007, declined slightly between 2008 and 2012, and thereafter exhibited a fluctuating upward trend, reaching a maximum in 2020. Column number concentrations remained relatively high with minor fluctuations, followed by a pronounced increase in 2020. Core size showed little variation (144–149 nm) and reached a minimum in 2018, while shell size exhibited pronounced fluctuations throughout the study period, hitting its lowest point in 2017. By contrast, South America experienced a rise in BC mass concentrations during 2005–2007, a decline in 2008–2009, and another increase in 2010, followed by sustained high but fluctuating levels. Number concentrations were elevated in 2005–2007, decreased between 2008 and 2012, and exhibited alternating downward and upward tendencies after 2013. Core size increased gradually from ~134 nm in 2005 to above 144 nm by 2012, followed by a decline and a subsequent recovery. Shell size stayed elevated during 2005–2015, with slight growth, but exhibited a fluctuating downward trend after 2016. In Europe, BC mass concentrations remained relatively stable, fluctuating slightly between 3000 and 4000 kg per grid per day from 2005 to 2020. Number concentrations also stayed within a stable range

with only minor variability. Core size peaked in 2016 (above 144 nm), while shell size fluctuated between ~680 and 690 nm without a monotonic trend. A similar pattern is observed in Africa, where both BC mass and number concentrations decreased slightly during 2005–2007, rose again in 2008, and subsequently remained at relatively high levels with modest variability. Core size stayed within 133–137 nm, showing only minor declines, while shell size remained around 687–692 nm without a discernible long-term trend. In East Asia, BC mass concentrations rose from 2005 to 2008 and then declined overall, reaching their lowest level in 2019. Number concentrations were relatively stable during 2005–2012 but decreased thereafter. Core size fluctuated between 127 and 133 nm, with a slight downward tendency. Shell size remained consistently between 695 and 700 nm throughout the period. For Southeast Asia, BC mass concentrations increased during 2005–2009, then declined and stabilized at lower levels. Number concentrations followed a similar trajectory, peaking during 2005–2009 before declining after 2010. Core size remained within 124–128 nm, showing little variation. Shell size showed gradual downward trends during 2005–2014 and 2015–2020. In South Asia, BC mass concentrations increased during 2005–2009, fluctuated between 3500 and 4000 kg per grid per day thereafter, and showed a distinct peak in 2018. Number concentrations followed a similar pattern, remaining high after 2009 and peaking in 2018. Core size declined overall, while shell size fluctuated between 707 and 712 nm without a monotonic trend. For North Asia, BC mass and number concentrations remained low and stable, with no significant long-term trends. Core size peaked in 2008 and then declined, stabilizing between 135 and 141 nm. Shell size showed limited variability, with a slight decreasing tendency and a minimum in 2008. Finally, Oceania exhibited increasing BC mass concentrations during 2005–2007, a decline from 2008 to 2012, and subsequent fluctuations within 2000–3000 kg per grid per day, with notable peaks in 2011 and 2017. Number concentrations showed a similar temporal pattern. Core size was more variable, and shell size displayed even greater fluctuations, with marked decreases in 2011 and 2017.

**Table 1. Annual average of BC atmospheric column mass concentration in 9 regions from 2005 to 2020.** Region codes are defined as follows: NA1 = North America, SA2 = South America, E3 = Europe, A4 = Africa, EA5 = East Asia, SEA6 = Southeast Asia, SA7 = South Asia, NA8 = North Asia, O9 = Oceania.

| BC MASS | NA1 | SA2 | E3 | A4 | EA5 | SEA6 | SA7 | NA8 | O9 |
|---|---|---|---|---|---|---|---|---|---|
| 2005 | 8549 | 5581 | 3210 | 6002 | 3160 | 3794 | 3314 | 1835 | 2269 |
| 2006 | 9624 | 6099 | 3450 | 6006 | 3386 | 4146 | 3608 | 2233 | 2504 |
| 2007 | 8991 | 7229 | 3357 | 5148 | 3252 | 3982 | 3350 | 1911 | 3649 |
| 2008 | 9529 | 5437 | 3028 | 5706 | 3695 | 4002 | 3914 | 2218 | 2334 |
| 2009 | 8456 | 5365 | 3313 | 5707 | 3296 | 4527 | 4152 | 1829 | 2593 |
| 2010 | 9467 | 6647 | 3206 | 5797 | 3470 | 3705 | 3549 | 2069 | 2354 |
| 2011 | 8470 | 5907 | 3007 | 5663 | 2942 | 4117 | 3553 | 1851 | 4892 |
| 2012 | 8101 | 6698 | 3125 | 5781 | 3193 | 3949 | 3792 | 2016 | 4039 |
| 2013 | 8063 | 5928 | 3245 | 5831 | 3343 | 3885 | 3627 | 1971 | 2246 |
| 2014 | 9215 | 5466 | 3250 | 5085 | 3142 | 3910 | 3291 | 1935 | 2661 |
| 2015 | 8035 | 5073 | 3329 | 5603 | 3179 | 3518 | 3590 | 2025 | 2375 |
| 2016 | 9620 | 6126 | 3849 | 5671 | 3241 | 3248 | 3657 | 1905 | 2202 |
| 2017 | 8728 | 5646 | 3723 | 5714 | 3024 | 3353 | 3550 | 2198 | 3435 |
| 2018 | 7410 | 5346 | 3263 | 5518 | 3189 | 3570 | 4325 | 1959 | 2556 |
| 2019 | 8501 | 6471 | 2905 | 5212 | 2834 | 3741 | 3426 | 1800 | 2355 |
| 2020 | 9717 | 6852 | 3803 | 5427 | 2922 | 3373 | 4039 | 1950 | 2176 |

## 3.3 Sensitivity analysis under different mixed states

Figure 5a presents time series of BC properties derived from different percentiles. For both column mass and number concentrations, the 50th percentile and the mean show nearly identical temporal patterns, particularly during high-concentration seasons (e.g., spring and summer), where peaks and fluctuations align closely. This consistency indicates that both metrics effectively capture background concentration trends and are well suited for monitoring long-term variability, with limited sensitivity to particle size assumptions. By contrast, the 75th percentile exhibits stronger variability and higher peaks during pollution episodes, reflecting enhanced sensitivity to high-concentration events. This makes the 75th percentile more appropriate for identifying extreme pollution episodes and rapid concentration increases. The 25th percentile remains comparatively stable with lower values and smaller fluctuations, providing reliable information on background low-concentration conditions.

For BC core and shell sizes, differences among percentiles are more pronounced. The 75th percentile consistently yields larger sizes than the 50th and 25th percentiles, particularly during periods of high concentration or large-particle events, with peaks notably higher than the mean. This demonstrates MISR's enhanced sensitivity to large BC particles when higher percentiles are applied, making the 75th percentile suitable for monitoring extreme conditions and large-size regimes. In

contrast, the 50$^{th}$ percentile and the mean display smoother temporal variations and similar overall trends, representative of typical mid-size particle distributions. The mean is slightly lower than the 50$^{th}$ percentile, suggesting that it integrates a broader range of particle sizes, including larger ones, and is thus suitable for characterizing overall distributions. The 25$^{th}$ percentile remains lowest, with minimal variability, emphasizing its relevance for representing small-particle conditions in relatively clean environments.

The PDFs (Fig. 5b) further illustrate these percentile-dependent sensitivities. For column mass and number concentrations, the 50$^{th}$ percentile and the mean show largely overlapping distributions, with similar frequency patterns, underscoring their robustness in representing general conditions. The 75$^{th}$ percentile shifts toward higher concentration ranges, with elevated frequencies, highlighting MISR's responsiveness to extreme pollution. The 25$^{th}$ percentile is concentrated in the low-concentration range, confirming its suitability for background monitoring. For BC core and shell sizes, the distributions diverge more strongly. The 25$^{th}$ percentile peaks at smaller size ranges, characterizing small-particle populations. The 50$^{th}$ percentile has a slightly broader distribution than the mean, covering more of the large-size range and reinforcing its representativeness under both background and polluted conditions. The 75$^{th}$ percentile is skewed toward much larger sizes, with substantially higher frequencies, indicating strong sensitivity to large-particle BC and supporting its use in high-pollution, large-size analyses. The mean overlaps largely with the 50$^{th}$ percentile, capturing overall particle size information across most conditions.

Overall, the use of multiple percentiles provides a layered characterization of MISR sensitivity. The 50$^{th}$ percentile and mean are most effective for describing general BC concentrations and typical size distributions, while the 75$^{th}$ percentile highlights high-concentration and large-particle events, and the 25$^{th}$ percentile emphasizes low-concentration and small-particle conditions. Together, these metrics offer complementary perspectives for refined monitoring of BC under diverse atmospheric environments.

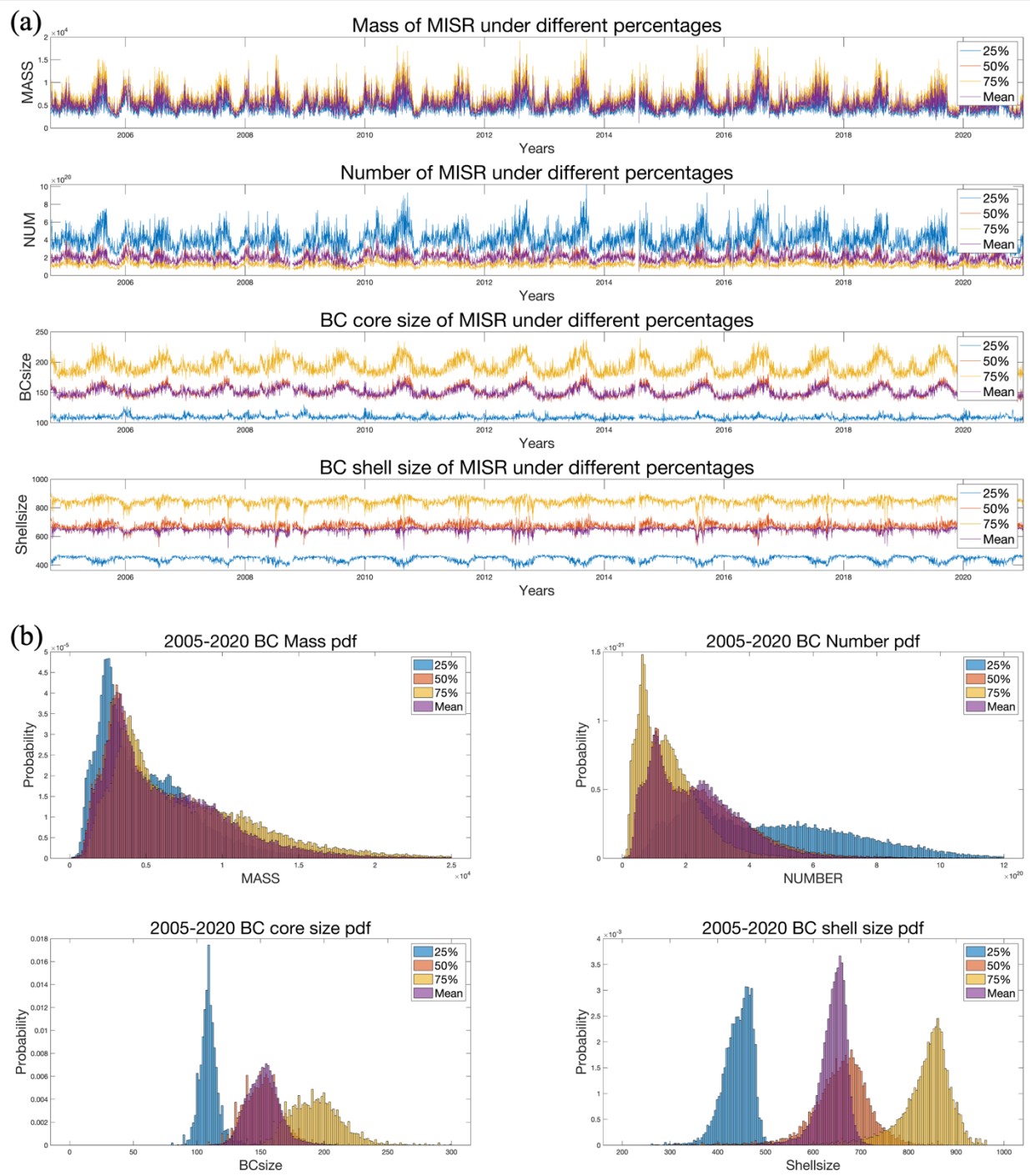

**Figure 5. Sensitivity analysis of various characteristics of BC under different percentages of core-shell sizes.** (a)Time series analysis, from top to bottom, including BC column mass concentration, column number concentration, core size, and shell size. (b) PDF analysis of BC mass concentration, column number concentration, core size, and shell size.

**3.4 Comparative Analysis of Black Carbon Observations: Insights from MERRA and MISR Datasets Across Global Regions**

Figures 6a and 6b illustrate that MISR and MERRA exhibit substantial discrepancies in the global distribution of BC column mass concentrations. MISR generally reports higher BC levels in Africa, South Asia, and East Asia, particularly over central-southern Africa, the Indian subcontinent, and both coastal and inland Chinese urban areas, whereas MERRA estimates are consistently lower. Only in limited regions, such as parts of Southeast Asia, does MERRA show relatively elevated signals, though it still tends to underestimate regional emissions overall. In South America, MISR detects enhanced

concentrations over the Amazon basin and southern regions, while MERRA shows weaker signals, indicating lower sensitivity to localized emission sources. In Europe and North America, both datasets reveal generally lower BC concentrations; however, MISR captures more localized pollution signals associated with industrial activity and wildfires, whereas MERRA remains comparatively smoother. In Oceania, MISR reports relatively higher concentrations in the north, while MERRA indicates higher values in the west.

Figure 6c presents the MISR-to-MERRA ratio, highlighting spatial differences between the datasets. Ratios exceeding unity dominate across much of Africa, South Asia, East Asia, and most of China, reflecting substantially higher BC estimates from MISR and its stronger sensitivity to local emission sources. Conversely, ratios near or below unity occur in northern South America, southern and central Australia, and parts of central and western Africa, suggesting higher MERRA estimates in those regions. Overall, MISR yields substantially higher ratios in major emission hotspots, enabling clearer identification of

localized pollution sources, whereas MERRA tends to emphasize background concentrations. In Europe, both datasets report relatively low BC, but their absolute values differ considerably: MISR captures greater spatial heterogeneity linked to industrial emissions and reports higher estimates overall. In North America, due to MISR's AOD > 0.25 threshold constraint, significant BC is retrieved primarily over fire-affected regions and eastern urban centers, with MISR values far exceeding MERRA, indicating that MERRA strongly underestimates episodic events. Over Oceania, where overall BC concentrations

are low, the two datasets show broadly consistent patterns across much of the central region; however, MERRA reports higher concentrations in the west (e.g., around Perth).

Figure 6d compares the PDFs of BC mass concentrations from the two datasets. MERRA exhibits a distribution concentrated in the low-concentration range, indicating its stronger capability in characterizing large-scale background conditions. By contrast, MISR shows higher frequencies in the elevated concentration range, highlighting its advantage in capturing

localized high-pollution events, especially over South America, South Asia, and East Asia–consistent with the spatial ratio patterns shown in Fig. 6c.

Figure 6e shows the MISR-to-MERRA ratio distribution under relatively low concentration conditions (MISR Mean25%). Compared with the mean ratio map, the overall ratios are closer to 1, indicating higher consistency between MISR and MERRA at lower MISR concentrations. However, in regions such as central–western Africa, southern Australia, and parts of

South America, areas with ratios below 1 become more pronounced, suggesting that MISR estimates are lower than those of MERRA. Over most of China, the ratio differences remain relatively small (within a factor of 1–2), reflecting a certain

degree of consistency. In northern Asia, the ratio declines, which may be related to MISR's detection sensitivity, the characteristics of local emission sources, and algorithmic uncertainties. Nevertheless, in northern Europe and South Asia, the MISR-to-MERRA ratios remain elevated, highlighting MISR's stronger sensitivity to localized pollution sources.

Figure 6f presents the ratio distribution under relatively high concentration conditions (MISR Mean75%). Most regions exhibit ratios significantly greater than 1, especially southern Africa, the Indian subcontinent, and East Asia, where MISR estimates far exceed those of MERRA. This indicates that MISR responds more strongly to BC in high-concentration regions, enabling clearer identification of emission sources and their intensities. In southwestern Australia and certain parts of western and central Africa, the ratios fall below 1, though the differences are limited in both spatial extent and magnitude. In

eastern China, India, and some high-emission areas of Southeast Asia, MISR values are particularly pronounced, reflecting the distinctive imprint of high-pollution events in its observations.

Figure 6g shows the ratio distribution based on the small-size assumption (MISR Size25%). Its spatial pattern closely resembles that in Fig. 6c, underscoring the stronger sensitivity of small-size BC concentrations to localized emissions. Because smaller BC particles have longer atmospheric lifetimes and can be transported over broader regions, they produce

stronger pollution signals in urban and industrial areas. Moreover, small-size BC is more responsive to meteorological conditions (e.g., temperature, humidity), resulting in more pronounced regional heterogeneity, particularly over parts of South America, Africa, and Asia.

Figure 6h presents the ratio distribution under the large-size assumption (MISR Size75%). Similar to Fig. 6c, its spatial distribution appears smoother, with relatively uniform ratios. This reflects the properties of large-size BC, which settles more

rapidly and has a shorter atmospheric lifetime, thus remaining concentrated near emission sources and producing more homogeneous large-scale distributions. Notably, the overall ratios in Fig. 6h are slightly lower than those in Fig. 6c, indicating that while MISR remains more responsive to local emissions than MERRA under the large-size assumption, the overall differences are smaller.

Taken together, Figs. 6e and 6f illustrate the differences between MISR and MERRA across concentration levels: MISR is

more sensitive in high-concentration regions, while agreement with MERRA is higher under low concentrations. Figs. 6g and 6h reveal contrasting behaviors under particle size assumptions: small-size BC better captures localized sources and transport features, whereas large-size BC is more spatially concentrated with smaller overall differences. These variations reflect the complexity of BC transport, deposition, and source impacts in the atmosphere. Overall, MISR demonstrates stronger sensitivity to high concentrations and localized sources, whereas MERRA more consistently reflects large-scale

background distributions.

In terms of global BC column concentration patterns, the two datasets show distinct characteristics. MERRA, being a model-based reanalysis product, exhibits relatively continuous and smooth spatial gradients. Across global regions, MERRA generally captures broad-scale concentration trends, particularly over high-BC regions in Africa, South America, Australia, and Asia, but with limited representation of localized details. In contrast, MISR, based on multi-angle imaging, provides

higher spatial resolution and stronger sensitivity to local variability, yielding richer spatial heterogeneity. The MISR-to-

MERRA ratio maps highlight that MISR estimates are substantially higher over regions such as sub-Saharan Africa and eastern South America, underscoring its responsiveness to strong emission sources. MISR also reveals more pronounced spatial variability within regions, particularly in high-emission areas such as South Asia, East Asia, and southern South America. This variability likely reflects the advantages of MISR's multi-angle observational approach, which captures
localized fluctuations and gradients in BC concentrations that MERRA tends to smooth over.

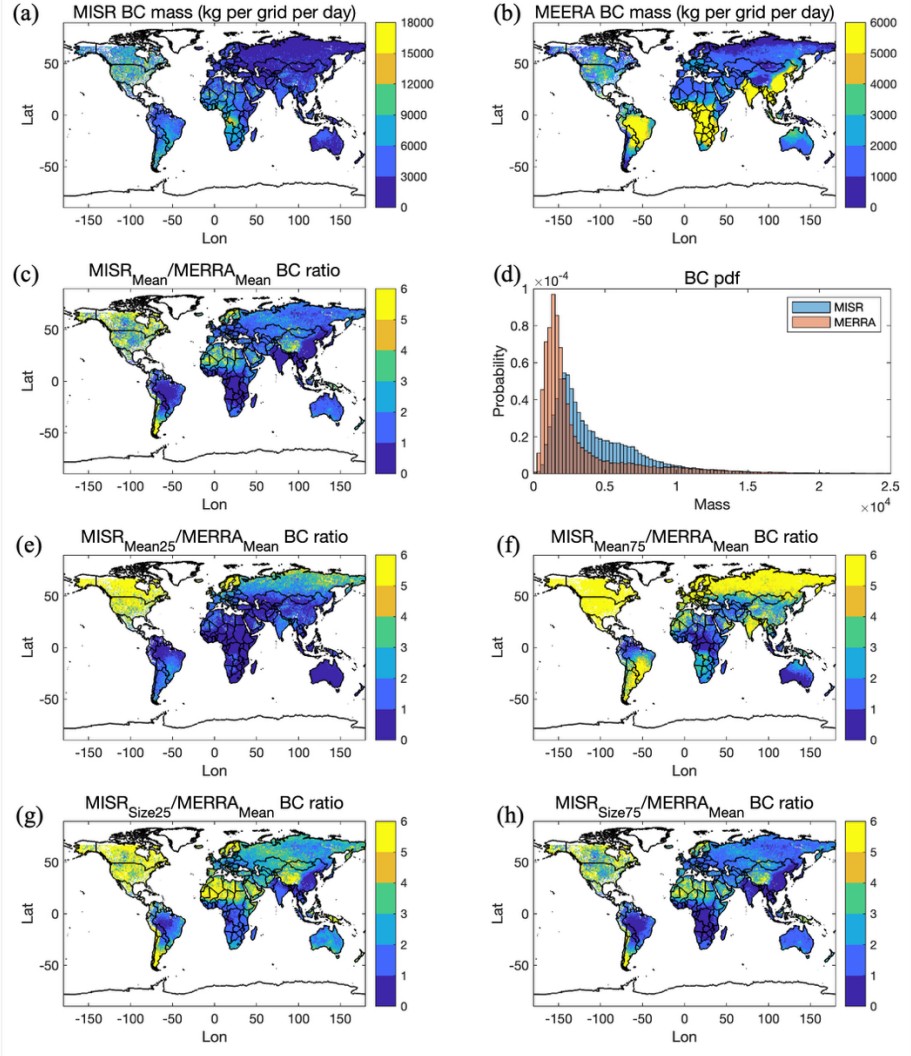

**Figure 6. Comparison between BC column concentration estimated based on MISR and MERRA data.** (a)MISR BC column mass concentration. (b)MERRA BC column mass concentration. (c)Ratio of mean BC column mass concentration between MISR and MERRA. (d)PDF of mean BC column mass concentration between MISR and MERRA. (e)Ratio of BC column mass concentration between MISR
(25% of mean) and MERRA (mean). (f)Ratio of BC column mass concentration between MISR (75% of mean) and MERRA (mean). (g) Ratio of BC column mass concentration between MISR (25% of size) and MERRA (mean). (h) Ratio of BC column mass concentration between MISR (75% of size) and MERRA (mean).

## 3.5 Distributional Patterns in Representative Regions

The spatial distribution and refinement process of BC products over South America are illustrated in Figure 7. The primary filtered regions are concentrated in central and southern South America, specifically within the Pantanal wetlands and the surrounding fringes of the Amazon Basin. As shown in Figure 7a, the temporal coverage criterion (valid observational days > 80%) retains the majority of these inland areas, ensuring the statistical representativeness of the annual cycle. Figure 7b displays the regions satisfying the BC column number concentration threshold (>70th percentile), effectively identifying the primary loading centers. The optical constraint applied in Figure 7c (AAOD$_{443/865}$, ratio < 8.0) further excludes potential dust-contaminated pixels, primarily along the arid fringes.

The overlapping result of these multiple constraints (Fig. 7d) yields a high-confidence sample set that captures the core of the South American biomass burning belt. Within these filtered regions, the BC column number concentration (Fig. 7e) and mass concentration (Fig. 7f) exhibit high spatial consistency, with elevated values corresponding to the intense fire activity zones identified in previous studies (Lin et al., 2020a; Wang et al., 2025). Furthermore, the retrieved microphysical properties provide additional detail, the BC particle size in these regions (Fig. 7g) and the corresponding shell thickness (Fig. 7h) demonstrate a clear spatial gradient from the fire source centers toward the downwind transport pathways. These filtered results capture high-mass samples characterized by significant inter-annual and intra-annual variations, reflecting the dynamic nature of biomass burning emissions in this region.

Figure 8 presents the filtered retrieval results over the Indian subcontinent, highlighting a different distribution regime. Unlike South America, the filtered regions in India are predominantly located in the southern and central states, specifically across Maharashtra, Tamil Nadu, and Karnataka (Fig. 8d). These regions are identified by simultaneously satisfying the temporal coverage (>70% valid days, Fig. 8a) and BC loading (>50th percentile, Fig. 8b) criteria, after excluding dust-prone areas using the spectral ratio constraint (AAOD$_{443/865}$ ratio < 7.0, Fig. 8c).

A key observation in the raw retrieval is that while the Indo-Gangetic Plain (IGP) and the Delhi National Capital Region exhibit extremely high BC column concentrations, they are largely excluded from the filtered dataset due to insufficient temporal coverage (valid observational days). In contrast, the filtered regions in southern and central India provide a more stable and persistent representation of the BC distribution. As shown in Figures 8e and 8f, BC number and mass concentrations in these states show strong spatial coherence. The retrieved microphysical parameters, including particle radii (Fig. 8g) and shell diameters (Fig. 8h), display signatures typical of mixed combustion sources. These microphysical traits are consistently observed across the identified hotspots, which align with areas characterized by high population density and diversified industrial activities. Although these regions exhibit lower peak concentrations compared to the seasonal extremes of the IGP, they represent long-term, sustained BC loadings that meet the rigorous data quality requirements of this study.

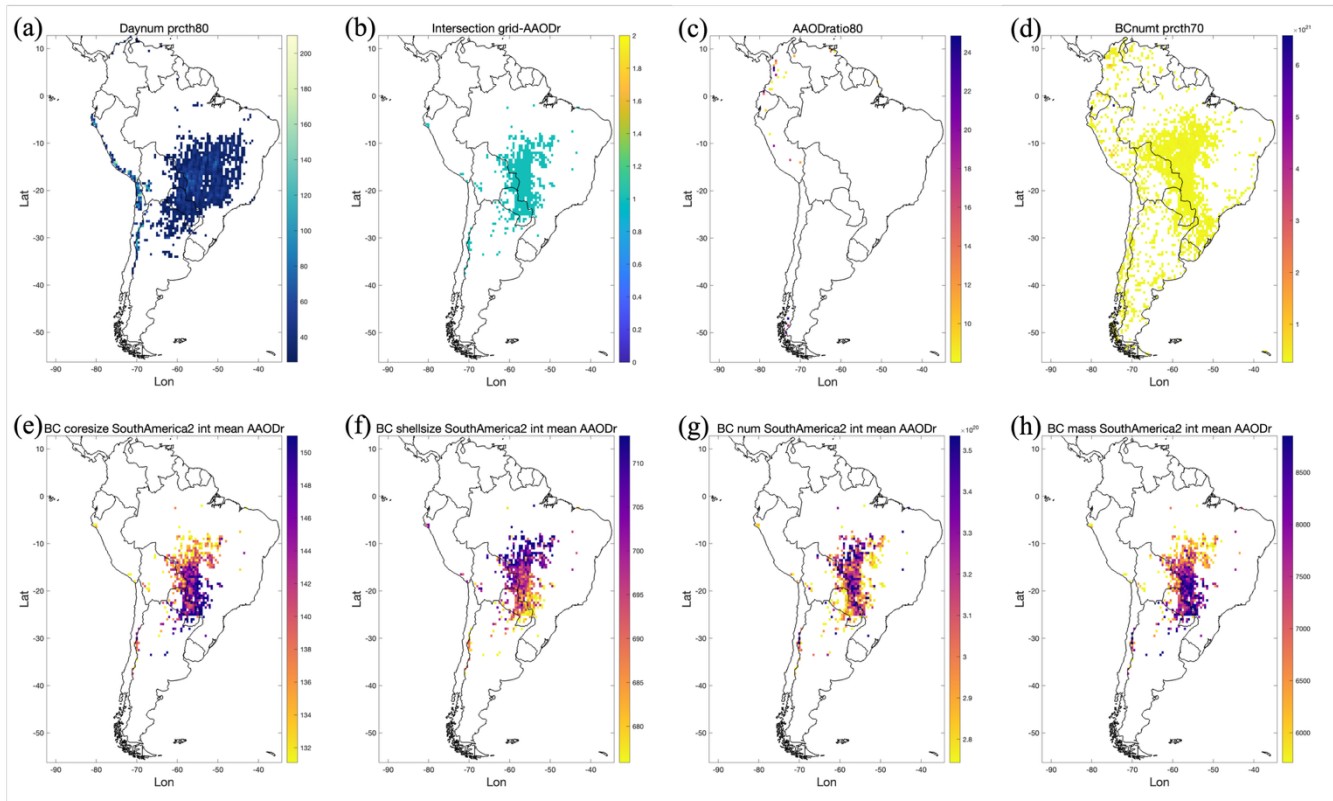

**Figure 7. Results of the analysis based on the BC column concentration product over South America.** (a)Regions with more than 80% valid observational days. (b)Regions with more than 70% BC column number concentration. (c)Regions excluded due to AAOD$_{443/865}$ values greater than 8.0. (d)Overlapping regions from (a) and (b) after excluding those regions identified in (c). (e)BC column number concentration in the filtered regions. (f)BC column mass concentration in the filtered regions. (g)BC particle size in the filtered regions. (h)BC Shell size in the filtered regions.

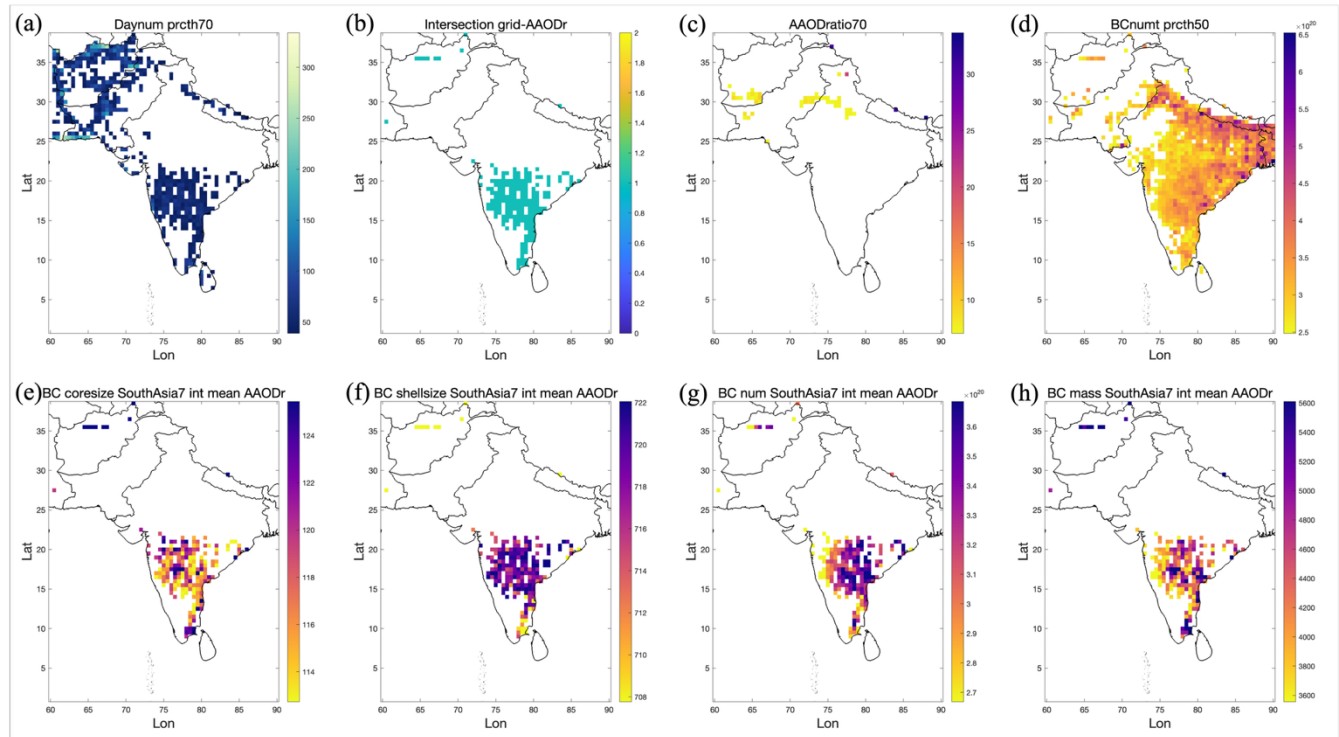

 **Figure 8. Results of the analysis based on the BC column concentration product over India.** (a) Regions with more than 70% valid observational days. (b) Regions with more than 50% BC column number concentration. (c) Regions excluded due to $AOD_{443/865}$ values greater than 7.0. (d) Overlapping regions from (a) and (b) after excluding those regions identified in (c). (e) BC column number concentration in the filtered regions. (f) BC column mass concentration in the filtered regions. (g) BC particle size in the filtered regions. (h) BC Shell size in the filtered regions.

## 4 Discussion

### 4.1 Physical Mechanisms and Climate Implications of Biomass Burning in South America

The spatial coherence of the filtered products over central and southern South America supports the interpretation that the applied constraints effectively capture a physically meaningful signal rather than scattered retrieval artifacts. The concentration of high BC mass loadings in the Pantanal wetlands and surrounding Amazon Basin coincides with regions where intense fire activity dominates during the dry season, reinforcing the role of biomass burning as a primary emission source (Krol et al., 2013). This consistency demonstrates that the retrieval framework accurately isolates fire-related signals (Freire et al., 2020), which remain robust even after AAOD-based screening to exclude mineral dust. Beyond the source areas, the influence of these emissions extends across vast spatial scales through the atmospheric processing of primary BC and co-emitted precursors. As smoke plumes are lofted and transported, they undergo in-situ aging, leading to the development of scattering coatings that enhance absorption via the lensing effect. These plumes can travel thousands of

kilometers to reach densely populated urban centers in Southeast Brazil, where long-range transport events severely deteriorate air quality and impose significant health risks (Santos et al., 2024).

From a climatic perspective, the identification of these BC hotspots reveals a tight coupling between fire activity and regional radiative perturbations. BC aerosols emitted from biomass burning can influence the South American climate system through semi-direct effects, where the absorption of solar radiation warms the aerosol layer and modifies atmospheric stability. This process typically reduces cloud cover and alters regional precipitation patterns and energy balances (Thornhill et al., 2018). The physical credibility of these findings is further validated by the long-term correspondence between our filtered products and AERONET monthly peaks at the CUIABA-MIRANDA station (Figure B1 and Table C4). Collectively, these results reproduce the canonical biomass-burning belts and emphasize the pivotal role of fire-climate feedbacks. Integrating such satellite-derived distributions with model simulations will be essential for future work to better quantify the long-term environmental impacts of biomass burning on the South American ecosystem.

## 4.2 Drivers of Persistent Anthropogenic Emissions in India

The filtered results over India reveal a distinct emission regime characterized by long-term stability rather than episodic extremes. While traditionally recognized hotspots like the IGP and Delhi exhibit high raw column concentrations, their exclusion due to insufficient temporal coverage suggests that the applied filtering approach favors capturing persistent emission patterns over isolated, high-value signals driven by sparse observations. The retained regions across Maharashtra, Tamil Nadu, and Karnataka represent leading industrial and economic states with high population densities and energy use (Shaban et al., 2022; Shanmugam and Odasseril, 2024; Ramachandran et al., 2025). In these southern filtered regions, the spatial agreement between BC concentrations and particle size metrics is consistent with domestic energy-use structures and industrial activities (Huang et al., 2023). In southern India, many households continue to rely heavily on traditional fuels such as firewood, crop residues, and dung cakes. Routine biomass burning in rural areas contributes to a sustained and stable source over these regions. These regions emissions are also strongly coupled with population distribution, leading to spatial BC patterns that align with population density. At the same time, industrial activity and urban emissions are also evident in these regions (Chathurangika et al., 2022). Particularly along industrial corridors and in the vicinity of large- and medium-sized cities, where the overlap of population-based combustion and coal-based industrial sources further amplifies local pollution levels. While the analysis in these regions is consistent with known conditions on the ground (Jaganathan et al., 2025; Kawano et al., 2025; Kunjir et al., 2025; Luo et al., 2025), these areas have generally been overlooked by global-scale emissions databases (Li et al., 2017; Kumar et al., 2023; Ren et al., 2025).

Importantly, this combined influence of biomass burning and industrial emissions in the filtered regions is not the result of singular extreme events but rather represents a persistent and widespread emission regime. Under the modulation of the monsoon system, such emissions are more easily transported and accumulated, thereby exerting prolonged and significant impacts on regional air mass when emissions are continuous (Wei et al., 2022). Unlike South America, where biomass burning is dominated by seasonal wildfires, BC emissions in central and southern India arise from a more complex

combination of household energy consumption, agricultural residue burning, and industrial production. This diversity of sources leads to environmental impacts that are correspondingly more intricate and persistent, shaping long-term regional air quality and monsoon-related climate feedbacks.

**4.3 Strengths, Limitations, and Recommended Use Cases of the Filtering Strategy**

The filtering strategy applied in this study prioritizes the capture of representative distributional features to ensure that the resulting products are statistically robust and regionally meaningful. However, this approach inherently introduces limitations regarding regions with limited observational samples but potentially extreme values remain areas of concern. For instance, the IGP and the surroundings of Delhi have long been regarded as the most prominent BC emission centers in India. Despite significant short-term aerosol loadings and their associated environmental risks, these areas were excluded from the present analysis due to insufficient valid observational days, often coupled with a high frequency of clean-air days during other periods of the year. While this filtering approach effectively highlights stable long-term patterns, it inevitably underrepresents certain hotspots driven by transient extreme pollution events or significant observation gaps. Therefore, studies focusing on short-term extremes or localized high-concentration impacts should combine raw observations with targeted monitoring strategies.

In summary, the multi-step filtering method successfully identified canonical biomass-burning regions in South America, where the spatial distribution, BC concentrations, and particle-size characteristics align with independent satellite observations and documented fire events. This validates the methodological framework and reveals the far-reaching impacts of biomass burning on air mass, regional climate, and public health. In contrast, the high-value regions identified in India, by contrast, highlight the combined effects of agricultural burning, industrial activity, and household biomass use, reflecting the long-term contributions of multiple overlapping sources. These regions are not only major drivers of air mass deterioration in South Asia but also key actors in climate feedbacks and health risks. Additional regional details and supplementary results, provided in the supporting materials and full dataset, offer further opportunities for in-depth regional analysis and thematic studies to extend understanding of BC's environmental impacts.

**5 Data availability**

The original input datasets used in this study, together with the derived BC column concentration products, and sample codes to access these datasets, are publicly available at Figshare with DOI: https://doi.org/10.6084/m9.figshare.30173917 (Liu et al., 2026). The dataset is primarily provided in MATLAB (.mat) format to preserve the multi-dimensional structure of the daily global and regional grids.

The data products are organized into two main categories, Global BC Products and Regional BC Products. The global products include gridded variables for column mass concentration (both per unit area and per grid point), column number concentration, BC core size, and shell size on a $0.5° × 0.5°$ resolution. The regional products provide high-resolution data for

nine predefined regions, including specific masks, valid observational day statistics, and filtered results based on optical constraints.

While this section provides a general overview, a more comprehensive description of the file hierarchies and variable definitions is provided in the Figshare repository. To facilitate immediate use, the repository also includes example MATLAB scripts for data loading, variable extraction, and spatial visualization. External auxiliary datasets, including MISR

aerosol products, AERONET CUIABA-MIRANDA field observations, and MERRA-2 reanalysis data, are also archived in the repository to ensure long-term accessibility and reproducibility.

## 6 Conclusions

This study developed an integrated methodology to estimate global BC column concentrations by combining MISR multi-band satellite retrievals with Mie scattering simulations under single scattering albedo constraints. Through a systematic

framework, we retrieved particle size distributions, absorption efficiencies, and subsequently derived BC column number and mass concentrations as well as two microphysical properties at a daily time step and global scale. Comparisons with AERONET and MERRA-2 datasets demonstrated the robustness and reliability of this approach across diverse regions and time periods. The results reveal distinct regional and temporal patterns of BC, highlighting the strong influence of biomass burning, industrial activity, and anthropogenic emissions in shaping both the magnitude and variability of BC concentrations.

The methodology significantly enhances the capacity to capture long-term and spatially extensive BC distributions, overcoming key limitations of traditional single-source observations. By providing a physically consistent and observation-driven dataset, this work offers valuable support for climate and air quality modeling, particularly in constraining radiative forcing estimates and assessing the environmental impacts of BC. Future applications of this approach can extend to other absorbing aerosols, improving predictive capability in Earth system models and informing targeted mitigation strategies at

both regional and global scales.

# 7 Appendix

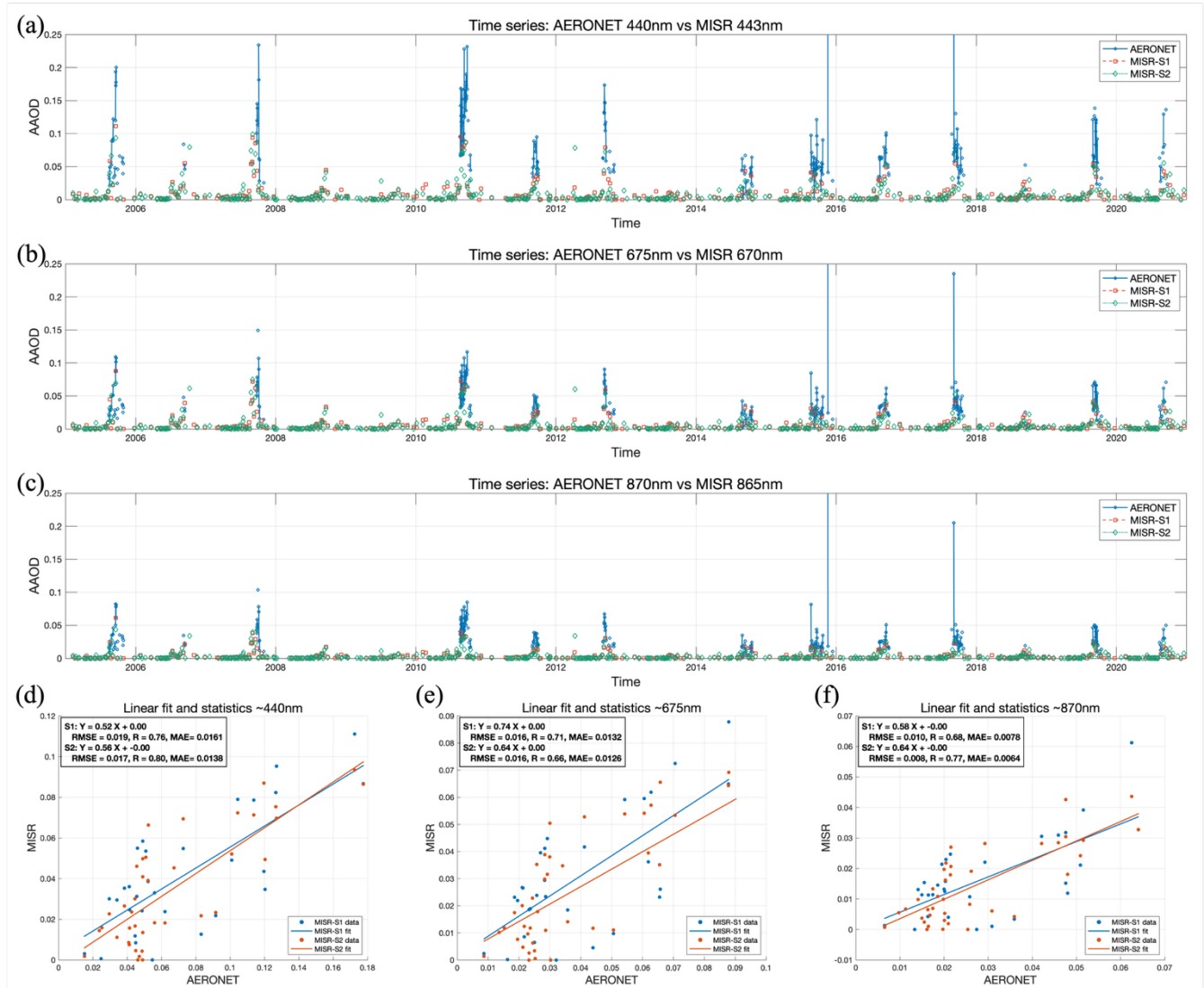

**Figure B1. Time series and correlation analysis of Absorption Aerosol Optical Depth (AAOD) comparing AERONET CUIABA-MIRANDA site observations (2005–2020) with two proximate MISR grid points (denoted as S1 and S2).** (a) AAOD at ~440 nm time series. (b) AAOD at ~675 nm time series. (c) AAOD at ~870 nm time series. (d) AAOD at ~440 nm correlation analysis (outliers removed). (e) AAOD at ~675 nm correlation analysis (outliers removed). (f) AAOD at ~870 nm correlation analysis (outliers removed).


**Table C1. Annual average of BC atmospheric column num concentration in 9 regions from 2005 to 2020.**

| BC NUM | NA1 | SA2 | E3 | A4 | EA5 | SEA6 | SA7 | NA8 | O9 |
|---|---|---|---|---|---|---|---|---|---|
| 2005 | 3.15E+20 | 2.69E+20 | 1.37E+20 | 2.60E+20 | 1.58E+20 | 2.28E+20 | 2.18E+20 | 7.63E+19 | 9.60E+19 |
| 2006 | 3.43E+20 | 2.62E+20 | 1.53E+20 | 2.58E+20 | 1.76E+20 | 2.38E+20 | 2.32E+20 | 9.10E+19 | 1.11E+20 |
| 2007 | 3.43E+20 | 3.04E+20 | 1.39E+20 | 2.37E+20 | 1.62E+20 | 2.58E+20 | 2.32E+20 | 7.65E+19 | 1.23E+20 |
| 2008 | 3.39E+20 | 2.34E+20 | 1.25E+20 | 2.48E+20 | 1.78E+20 | 2.55E+20 | 2.42E+20 | 8.25E+19 | 9.83E+19 |
| 2009 | 3.18E+20 | 2.27E+20 | 1.42E+20 | 2.49E+20 | 1.70E+20 | 2.72E+20 | 2.84E+20 | 7.47E+19 | 1.09E+20 |
| 2010 | 3.41E+20 | 2.84E+20 | 1.45E+20 | 2.62E+20 | 1.81E+20 | 2.27E+20 | 2.55E+20 | 8.86E+19 | 1.03E+20 |
| 2011 | 3.25E+20 | 2.36E+20 | 1.34E+20 | 2.63E+20 | 1.51E+20 | 2.34E+20 | 2.45E+20 | 7.72E+19 | 1.78E+20 |
| 2012 | 3.18E+20 | 2.62E+20 | 1.36E+20 | 2.66E+20 | 1.68E+20 | 2.34E+20 | 2.49E+20 | 8.65E+19 | 1.42E+20 |
| 2013 | 3.23E+20 | 2.33E+20 | 1.45E+20 | 2.67E+20 | 1.75E+20 | 2.18E+20 | 2.56E+20 | 8.51E+19 | 9.87E+19 |
| 2014 | 3.23E+20 | 2.44E+20 | 1.34E+20 | 2.45E+20 | 1.65E+20 | 2.19E+20 | 2.41E+20 | 8.32E+19 | 1.07E+20 |
| 2015 | 3.16E+20 | 2.35E+20 | 1.45E+20 | 2.66E+20 | 1.63E+20 | 2.30E+20 | 2.46E+20 | 8.80E+19 | 9.71E+19 |
| 2016 | 3.49E+20 | 2.48E+20 | 1.41E+20 | 2.70E+20 | 1.61E+20 | 1.86E+20 | 2.61E+20 | 8.25E+19 | 9.89E+19 |
| 2017 | 3.30E+20 | 2.42E+20 | 1.52E+20 | 2.76E+20 | 1.53E+20 | 1.98E+20 | 2.52E+20 | 8.54E+19 | 1.36E+20 |
| 2018 | 3.07E+20 | 2.25E+20 | 1.32E+20 | 2.64E+20 | 1.65E+20 | 1.97E+20 | 3.02E+20 | 8.00E+19 | 1.06E+20 |
| 2019 | 3.19E+20 | 2.60E+20 | 1.22E+20 | 2.56E+20 | 1.45E+20 | 2.07E+20 | 2.64E+20 | 7.64E+19 | 1.03E+20 |
| 2020 | 3.83E+20 | 2.56E+20 | 1.48E+20 | 2.60E+20 | 1.45E+20 | 2.01E+20 | 2.54E+20 | 7.74E+19 | 9.47E+19 |


**Table C2. Annual average of BC core size in 9 regions from 2005 to 2020.**

| CORE SIZE | NA1 | SA2 | E3 | A4 | EA5 | SEA6 | SA7 | NA8 | O9 |
|---|---|---|---|---|---|---|---|---|---|
| 2005 | 148 | 134 | 136 | 138 | 131 | 126 | 121 | 138 | 136 |
| 2006 | 150 | 139 | 138 | 137 | 130 | 127 | 122 | 140 | 137 |
| 2007 | 146 | 142 | 139 | 135 | 133 | 124 | 121 | 141 | 146 |
| 2008 | 150 | 139 | 140 | 138 | 135 | 124 | 125 | 148 | 138 |
| 2009 | 147 | 139 | 138 | 137 | 131 | 127 | 119 | 140 | 137 |
| 2010 | 150 | 141 | 134 | 135 | 130 | 127 | 119 | 137 | 136 |
| 2011 | 145 | 143 | 136 | 134 | 130 | 127 | 118 | 140 | 144 |
| 2012 | 145 | 144 | 136 | 135 | 130 | 127 | 121 | 138 | 145 |
| 2013 | 144 | 142 | 136 | 135 | 128 | 127 | 118 | 136 | 135 |
| 2014 | 151 | 136 | 138 | 134 | 127 | 128 | 117 | 135 | 138 |
| 2015 | 145 | 135 | 137 | 133 | 128 | 125 | 120 | 136 | 139 |
| 2016 | 149 | 142 | 144 | 133 | 129 | 128 | 117 | 136 | 133 |
| 2017 | 149 | 139 | 140 | 133 | 129 | 127 | 119 | 140 | 141 |
| 2018 | 142 | 140 | 139 | 133 | 128 | 128 | 119 | 139 | 139 |
| 2019 | 147 | 142 | 138 | 132 | 128 | 127 | 116 | 138 | 138 |
| 2020 | 145 | 145 | 142 | 133 | 131 | 127 | 120 | 141 | 136 |

**Table C3. Annual average of BC shell size in 9 regions from 2005 to 2020.**

| SHELL SIZE | NA1 | SA2 | E3 | A4 | EA5 | SEA6 | SA7 | NA8 | O9 |
|---|---|---|---|---|---|---|---|---|---|
| **2005** | 673 | 697 | 691 | 688 | 696 | 702 | 709 | 690 | 685 |
| **2006** | 665 | 689 | 664 | 686 | 699 | 698 | 710 | 688 | 689 |
| **2007** | 674 | 690 | 683 | 691 | 698 | 699 | 711 | 688 | 660 |
| **2008** | 664 | 695 | 678 | 689 | 696 | 698 | 709 | 655 | 681 |
| **2009** | 676 | 686 | 685 | 688 | 698 | 696 | 712 | 685 | 688 |
| **2010** | 665 | 693 | 688 | 691 | 698 | 697 | 710 | 691 | 679 |
| **2011** | 681 | 690 | 684 | 691 | 701 | 696 | 711 | 685 | 644 |
| **2012** | 677 | 691 | 689 | 687 | 699 | 694 | 709 | 689 | 652 |
| **2013** | 679 | 685 | 680 | 689 | 697 | 698 | 707 | 689 | 687 |
| **2014** | 665 | 697 | 680 | 692 | 701 | 692 | 713 | 693 | 688 |
| **2015** | 671 | 698 | 685 | 687 | 697 | 700 | 708 | 685 | 680 |
| **2016** | 658 | 689 | 680 | 689 | 698 | 696 | 708 | 692 | 691 |
| **2017** | 646 | 691 | 686 | 691 | 699 | 696 | 710 | 686 | 645 |
| **2018** | 674 | 690 | 684 | 689 | 698 | 693 | 708 | 685 | 673 |
| **2019** | 668 | 686 | 683 | 692 | 700 | 692 | 712 | 687 | 691 |
| **2020** | 674 | 678 | 669 | 691 | 696 | 693 | 708 | 685 | 689 |


**Table C4. Statistical performance metrics of MISR-derived AAOD against AERONET observations at the CUIABA-MIRANDA site (2005–2020).** *nRMSE* and *nMAE* denote *RMSE* (Root Mean Square Error) and *MAE* (Mean Absolute Error) normalized by the mean of AERONET observations, respectively.

| BAND | RMSE_S1 | MAE_S1 | NRMSE_S1 | NMAE_S1 | RMSE_S2 | MAE_S2 | NRMSE_S2 | NMAE_S2 |
|---|---|---|---|---|---|---|---|---|
| 1 | 0.019 | 0.016 | 45% | 39% | 0.017 | 0.014 | 47% | 39% |
| 2 | 0.016 | 0.013 | 51% | 43% | 0.016 | 0.013 | 62% | 49% |
| 3 | 0.010 | 0.008 | 57% | 46% | 0.008 | 0.006 | 54% | 44% |

## 8 Author contributions

JBC was responsible for conceptualization and funding acquisition, and also supervised the study. ZWL and JBC conducted the formal analysis and validation. ZWL and JBC, together with LYG and SW, developed the methodology. ZWL performed the software development, data processing, and visualization. ZWL and JBC carried out the investigation, with additional resources provided by SW, ZQL and ZWL. The original draft was prepared by ZWL, JBC, and PT, while JBC, ZWL, PT, and KQ contributed to review and editing.

## 9 Competing interests

The authors declare that they have no conflict of interest.

## 10 Acknowledgements

We acknowledge the MISR science team for providing the MISR products used in this study. We also acknowledge the
AERONET principal investigator and staff at the CUIABA-MIRANDA site for maintaining and providing high-quality
ground-based aerosol observations. In addition, we acknowledge the NASA Global Modeling and Assimilation Office
(GMAO) for providing the MERRA-2 reanalysis data.

## 11 Financial support

This work was supported by the Natural Science Foundation of Jiangsu Province (Grants No BZ2024060).

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
