# Peer review of "A Global Black Carbon Dataset of Column Concentration and Microphysical Information Derived from MISR Multi-band Observations and Mie Scattering Simulations"

_Earth System Science Data, 2025_

## Author Comment (AC1)

**Dear Editor and Reviewers**

We sincerely thank both reviewers for their insightful and constructive comments. We have carefully revised the manuscript to address all concerns. Below we provide a point-by-point response, with the reviewers' unedited comments highlighted in yellow, and our responses are highlighted in blue. All revisions have been incorporated into the updated manuscript, with major modifications emphasized in the responses and *specific textual changes shown in italics*.

We thank the reviewers again for their constructive suggestions. We believe that the extensive revisions—including restructuring, improved methodology descriptions, full dataset documentation, expanded uncertainty analysis, and clearer scientific justification—have significantly strengthened the manuscript and made it suitable for publication in *Earth System Science Data*.

We hope the revised manuscript satisfactorily addresses all concerns. If there are any further questions or comments, please let us know.

Kind Regards,

Jason Cohen

**Reviewer #1 comments**

**General Comments:**

Liu et al. processed 16 years of MISR Level-3 aerosol products to derive global black carbon (BC) information, including its column concentration and microphysical properties, based on a theoretical Mie model. The authors developed the new dataset to support investigations of the spatial and temporal variations of BC concentrations and emissions at both global and regional scales. Comparisons with AERONET observations and MERRA-2 model products are valuable additions that help demonstrate the dataset's performance.

**Specific comments:**

However, several major issues need to be addressed before the manuscript can be considered for publication, even prior to providing more specific comments.

Overall, the current manuscript compiles a large amount of information without clear organization or illustration, which significantly reduces its readability and scientific value. Therefore, the paper requires substantial restructuring to clearly separate each section.

At present, much of the introductory and methodological content is misplaced within the Results section, while the Discussion section mixes results with interpretation.

Recommend reorganizing the manuscript so that all methodological descriptions are moved to the Methods section, and all results are presented in the Results section, and relevant interpretations and implications are discussed in the Discussion section.

We sincerely appreciate the reviewer's insightful remarks regarding the need for substantial restructuring of the manuscript and a visual graphic to aid in understanding. We agree that the previous version lacked the organizational clarity that you expected, and aim to meet your standard. Following this suggestion, we have reorganized the manuscript to clearly separate Methods, Results, and Discussion in line with your suggested conventions.

In the revised version, all methodological descriptions have been consolidated and expanded into a dedicated Section 2 (Materials and Methods), ensuring that all technical details, assumptions, and computational steps are presented in a coherent and logically consistent manner. The Results section has been rewritten to focus on dataset characteristics, spatial and temporal patterns of the resulting Black Carbon [BC] column and microphysical properties, and key features relevant to those who download and/or use the dataset, avoiding methodological overlap with earlier sections. The Discussion section has been reshaped to emphasize interpretation of the dataset, regional considerations, and limitations, rather than repeating descriptive results.

Furthermore, we have added in a flowchart as Figure 2, with the goal of making the processes used in this work easier to follow.

Collectively, these revisions significantly enhance readability and present the dataset in a clearer and more accessible format.

The authors can also follow the structure and presentation style of other published ESSD data papers to effectively demonstrate their dataset and provide illustrative examples.

We appreciate the reviewer's suggestion to follow the structure and presentation style commonly used in many ESSD data papers. As noted in our earlier response, we have already carried out a comprehensive reorganization of the manuscript, and hope that it now structurally conforms to your expectations.

The revised manuscript now follows flow from Introduction through Methods, Results, Discussion, and the data availability statements, and this restructuring has greatly improved the clarity and consistency of the presentation.

In addition, we have substantially expanded the dataset description, including detailed explanations of file structure, variable definitions, data formats, and usage guidance. These additions directly address the reviewer's concern regarding effective dataset demonstration.

Further elaboration on data accessibility and illustrative examples is provided in our response to the final comment, where we describe the newly added Dataset Description section in detail. Together, these revisions ensure that the manuscript follows the organizational and stylistic expectations of ESSD and presents the dataset in a clear, well-documented, and user-oriented manner.

The Methodology section, needs significant improvement. In addition to relocating relevant content from the Results section, the authors should provide detailed explanations of the equations used, explicitly listing and defining all key variables. Without these details, the equations appear as disconnected expressions without clear purpose or interpretability.

We thank the reviewer for these valuable comments regarding the clarity and completeness of the Methodology section. We fully agree that the previous version did not sufficiently explain the equations and variables used in the inversion, and that the earlier subsection describing classical Mie theory (Lines 135–150) was overly general. In response, we have made extensive revisions to the entire Methodology section. First, the generic and textbook-like description of Mie theory has been substantially reduced, and the original equations in that subsection have been removed, as suggested by Reviewer 2. Instead, we focused the section on the specific inversion framework developed in this study, its underlying assumptions, and the computational steps by which MISR SSA and AAOD observations across different wavelengths and associated uncertainty ranges are used in tandem to constrain BC microphysical properties.

To address the concerns raised by Reviewer 1, we have rewritten the methodology to provide detailed and explicit explanations of all equations used in the inversion. Every variable now appears with a precise definition, physical meaning, and unit, ensuring that equations no longer appear as isolated expressions but as fully interpretable components of the retrieval algorithm. We also clarified the main parameter settings and assumptions used in the core-shell Mie simulations (e.g., prescribed complex refractive indices and density for both core and shell, as well as variable core and shell size ranges—meaning variable mixing states), as well as how we employ a constrained solution using the individual wavelength-dependent constraints from MISR's four

spectral bands, to provide the necessary context for interpreting the equations within the full computational workflow.

Collectively, these revisions improve the transparency and interpretability of the Methodology section and support reproducibility, while keeping the presentation focused on the information most needed to understand and apply the dataset.

Furthermore, the procedures for spatial and temporal aggregation of the MISR product should be clearly described, and the QA/QC processes for the MISR L3 aerosol data must be clarified, including an assessment of associated uncertainties. The uncertainties associated with AERONET and MERRA-2 estimates should also be carefully discussed.

We thank the reviewer for these detailed and technical suggestions. We agree that a transparent description of the data processing chain and a robust discussion of uncertainties are essential. In fact, we believe that this philosophical approach guides our entire work approach, so we are very happy to address this more explicitly. We have addressed these points by incorporating official MISR technical specifications and established literature:

1. Spatial and Temporal Aggregation of MISR Product:

In the revised manuscript (Section 2.1), we have clarified that our study utilizes the official MISR Level 3 Component Global Aerosol (CGAS) product. This product provides global summaries of higher-resolution (4.4 km × 4.4 km) MISR Level 2 aerosol retrievals onto a 0.5° × 0.5° geographic grid. According to the MISR Data Product Specification, these grid cell averages are calculated by assigning equal weight to every valid Level 2 sample within the cell, regardless of temporal sampling frequency. While this resampling may yield some representation errors, especially on grids which have intense sources over a small subset of the grid, it seems to be a commonly adapted approach even with harder to retrieve substances such as $CH_4$ (Chen et al., 2023), meaning that its longer-term application with species far easier and cleaner to retrieve like AOD and SSA, should pose fewer problems. Already our values of BC number loading are relatively higher than reanalysis and our SSA is relatively lower

than reanalysis, and this daily spatially aggregated product should help to eliminate the most extremely high loading and extremely low SSA values that would arise if we were to use the original 4.4km x 4.4km data, consistent with differences observed over Bangladesh when we used the lower resolution OMI (Liu et al., 2024a) as compared with the higher resolution TROPOMI (Tiwari et al., 2025).

2. QA/QC and Uncertainties for MISR L3 Data:

We have updated and expanded the description of the built-in QA/QC information provided with the official MISR products. Only Level 2 retrievals that pass all internal quality tests—specifically those with "Aerosol_Retrieval_Screening_Flags" set to 0—are included in the Level 3 summary calculations. Because we use the official, mature CGAS Level 3 product, we do not repeat the full set of internal QA/QC procedures in detail; instead, we clarify which quality screening is inherited from Level 2 and cite the relevant product documentation and validation literature. As a result, regions with typically poor retrieval quality, such as Greenland and Antarctica, are excluded to ensure data reliability. The CGAS product inherits a "Stage 3 Validated" maturity level, and its expected precision is consistent with the established performance of the Level 2 algorithm (e.g., AOD uncertainty of ±0.05 or 20% × AERONET, whichever is larger).

3. Uncertainties Associated with AERONET and MERRA-2:

We have added a dedicated discussion on the inherent uncertainties of our comparison targets. We now explicitly state the typical uncertainty ranges for AERONET (e.g., from -0.02 to +0.06 for AOD, ~0.03 for SSA) (Giles et al., 2019; Sinyuk et al., 2020). We further discuss how MERRA-2 uncertainties are mostly linked to emission inventory limitations (Li et al., 2025; Liu et al., 2024b), which provides critical context for the regional discrepancies observed in our analysis.

A detailed comparison between MISR, MODIS, and AERONET over China has yielded that MISR has both higher correlation coefficient and lower difference when compared with AERONET compared with MODIS. Although the uncertainty does still fall outside of the AERONET uncertainty range, indicating a low bias under high AOD conditions, the difference is still found to be within the MISR uncertainty range (Kahn

et al., 2010; Garay et al., 2020; Si et al., 2020). Anecdotally, an older paper by the corresponding author using MISR (Cohen, 2014) was able to pick up a substantial signal of absorbing aerosols over the Maritime Continent, while newer work by the corresponding author using OMI (Liu et al., 2024a) was not able to do so, supporting the point that MISR may be the best suited individual satellite for the tasks set forth in this work.

Finally, the description of the new dataset should be more comprehensive. The authors need to clearly specify the data structure, file format, and accessibility. For example, if the dataset is mainly provided in ".mat" format, they should offer example MATLAB scripts or guidance for reading and visualizing the data.

We thank the reviewer for pointing out the need for a more comprehensive and user-oriented description of the dataset. In the revised manuscript, we have substantially expanded the dataset description to clearly present its structure, content, and accessibility. We now explicitly describe the spatial grids stored in the files, the key BC variables, and the associated observational inputs. These clarifications ensure that users can easily understand how the dataset is organized and how the different variables relate to one another. A more comprehensive description of the data product is now available via the Figshare repository link, where users can find detailed information on the file hierarchy and variable definitions.

To further enhance accessibility and practical usability, we have also provided example MATLAB scripts that demonstrate how to load the ".mat" files, extract individual variables, and generate visualizations of the BC column concentration fields. These examples offer readers a direct and reproducible approach for exploring the dataset without requiring additional preprocessing. Together, these improvements offer a clearer and more complete description of the dataset's structure and use, fully addressing the reviewer's concerns and aligning with ESSD's expectations. They also increase the usability of the data product for the broader scientific community.

Overall, I recommend that the authors substantially revise and better structure the manuscript before it can be further considered for publication.

**Reviewer #2 comments**

**General Comments:**

This study by Liu et al. described an integrated method to estimate global BC mixing state, column concentration by using MISR VIS-NIR spectral SSA products with MIE calculation. The comparison with AERONET and MERRA-2 data show reliability of the developed method. The manuscript mainly focuses on the discussion about the >15 years global BC column concentration results. Overall, the topic is interesting. However, in my view, the manuscript still needs substantial improvements before it can be considered for publication.

The main concerns are as follows:

1. The introduction and data description sections lack appropriate context and references. For example, Section 2.1 presents the MISR data on which the study heavily relies, yet it does not include a single reference. In addition, the description of MIE scattering in Section 2.2 is presented at a very general level and without any proper references. I don't see any connection with the method used in this study.

We thank the reviewer for highlighting the lack of contextual information and missing references in the Introduction and in the early parts of the Data and Methods section. We agree that these omissions weakened the clarity of the manuscript and did not sufficiently situate the study within existing work.

In the revised version, we have added essential references throughout the introduction as well as Sections 2.1 and 2.2, including citations describing the MISR instrument, the documentation for the MIL3DAEN product on which this study relies, as well as various works describing the uncertainties in the MISR products and comparisons with AERONET and other surface networks. These additions now provide the appropriate scientific background and acknowledge the foundational work underpinning the MISR dataset. As of this revision, we now include 14 references directly related to MISR.

Similarly, Section 2.2 has been substantially improved. The earlier generic description of Mie scattering, which indeed lacked clear relevance to the proposed method, has been removed and replaced with a concise explanation focused directly on the specific Mie computations, parameters, and assumptions used in our inversion framework. A more detailed explanation of how this revision was implemented is provided in our response to the reviewer's subsequent comment.

Overall, the updated section now clearly explains the inputs, physical assumptions, and role of SSA/AAOD constraints within the retrieval method rather than repeating textbook material. These revisions collectively provide a clearer scientific context, correct the missing-citation issue, and make the methodological narrative more coherent and relevant to the goals of this study.

2. The most critical issue lies in the methodology section: the manuscript lacks a detailed description and evaluation of the methods used. As a result, the subsequent discussion of global BC column concentrations is not grounded on a sufficiently robust basis. For example, the refractive index of BC can significantly affect the derived column concentrations; however, the manuscript does not specify the key parameters used, such as the refractive index or particle density.

We thank the reviewer for drawing attention to this critical issue in the original methodology section. We fully agree that the previous description did not provide sufficient detail to support a rigorous understanding of the inversion framework, nor did it adequately specify the key physical parameters that directly influence the retrieved BC column concentrations.

Recognizing that the refractive index, particle density, and microphysical assumptions are central to the interpretation of results, we have substantially revised the methodology to include a comprehensive and fully transparent description of these elements. The revised manuscript now clearly states the wavelength-dependent refractive indices adopted for BC (Schuster et al., 2005), and specifies the BC material density of 1.8 g cm$^{-3}$ (Bond and Bergstrom, 2006) as well as the density and compositional assumptions applied to the coating material. We also provide a detailed description of the fact that we use a method of a flexible size distribution,

actively examining all possible particle size solutions at 10nm resolution of both the BC and the shell coating material in the optical computations, including the full range of core and shell radii employed in the inversion (Wang et al., 2021a; Tiwari et al., 2023). Furthermore, we clarify how MISR's four-band SSA constraints are used in a method to ensure that the optical properties of modeled particles remain consistent with the spectral absorption and scattering characteristics observed by the instrument across all 4 wavebands in tandem (Liu et al., 2024), not only using one waveband and some hypothetical angstrom or absorbing angstrom exponent (Tiwari et al., 2025), which provides a highly non-linear and realistic filter, leading to a far closer match with surface observations (Guan et al., 2025).

These additions not only offer a more robust physical foundation for the retrieval but also make the entire inversion process reproducible and scientifically interpretable. By explicitly documenting all essential inputs, assumptions, and parameter choices, the methodology now provides the rigor necessary for supporting the subsequent discussion of global BC column concentrations.

3. In accordance with ESSD guidelines, the authors should include the generated dataset and its registered DOI in the abstract. Furthermore, the main text must provide a general overview of the data structure.

We thank the reviewer for underscoring the importance of clearly identifying the dataset within both the abstract and the main text, in accordance with ESSD guidelines.

During the initial editorial screening, we were asked to remove the DOI from the abstract for formatting reasons, but following this reviewer's recommendation we have now reinstated the dataset DOI so that readers can immediately access the data product.

In addition, the revised manuscript provides a more explicit overview of the dataset in the main text. We expanded the relevant sections to give a concise but informative description of the dataset's organization, including the arrangement of spatial grids, the principal retrieved variables, and the way these elements are stored within the

distributed ".mat" files. These additions ensure that readers can readily understand the structure and content of the dataset without consulting external resources. Further specifics regarding the file hierarchy and variable definitions of the data product have been updated in the Figshare repository, which users can access for more detailed information.

Together, the reintroduced DOI in the abstract and the strengthened dataset overview in the manuscript enhance transparency, improve discoverability, and align the paper with ESSD's expectations for data accessibility and documentation.

**Specific comments:**

L12: Multi-angle Imaging SpectroRadiometer (MISR)

We thank the reviewer for noting that the abbreviation "MISR" was missing after the full instrument name. To improve clarity and conform to standard scientific notation, we have added the abbreviation "*(MISR)*" following the first occurrence of "Multi-angle Imaging SpectroRadiometer" in the manuscript. This correction has been implemented at Line 15, and we have reviewed the surrounding text to ensure consistent use of the full name and its abbreviation throughout the manuscript. We appreciate the reviewer's attention to this detail, which helps enhance the readability and precision of the paper.

L25-27: this statement is debatable according to the latest IPCC AR6

We would like to express our sincere gratitude to the reviewer for pointing out the nuance regarding the radiative forcing of black carbon (BC) as stated in the IPCC AR6. We acknowledge that the exact ranking and magnitude of BC's climate forcing remain a subject of active scientific debate. However, our statement is based on a growing body of evidence of over 30 papers, including but not limited to Cohen and Wang (2014), Mann et al. (2010), Chen et al. (2022), Everett et al. (2022), Kelesidis et al. (2022), Sedlacek et al. (2022), Ramachandran et al. (2023)—which suggests that many models integrated into the IPCC framework, such as the UKESM1, may underestimate BC's emissions and/or column loading when based on AAOD observations, hence leading to vast changes in its impact. These models often rely on simplified, fixed assumptions

regarding particle size distributions and mixing states (e.g., the lensing effect from core-shell structures), which may not fully capture the complexities of high-emission regimes. It is important to understand that the IPCC AR6 was supposed to be a comprehensive summary, and it is odd that upwards of 30 papers which all mentioned this were continently left out. For this reason, we do not choose to state the IPCC AR6 results, and instead prefer to refer to the published results across the community.

The primary objective of our study is to provide an observationally constrained global dataset that "patches the hole" left by traditional model parameterizations and simplifications, as well as missing or spatially or temporally misplaced emissions sources (Lu et al., 2025; Wang et al., 2021b, 2025). By integrating MISR multi-angle observations with a Mie scattering framework, we offer a more nuanced representation of BC's physical and optical properties across different regions. We do reference some of the underlying papers which support the view as summarized by the AR6 as well. However, we also point out that the vast majority of these papers are not compared with AAOD or SSA observations, and do not generally match well with them. We therefore have produced this work as a means to provide observationally based constraints on the loading and microphysical properties of BC. We hope that the modeling and remote sensing communities both may use this as a way to better nudge their models to match better with AAOD and SSA based observations as well as to provide a more realistic starting point for inversions.

We have revised the text in Lines 29-36 as follows:

"*BC's strong solar radiation absorption capacity makes it potentially among the most critical climate forcing agents after carbon dioxide. The exact magnitude of its radiative impact remains a subject of ongoing debate in recent assessments (e.g., IPCC AR6) as well as many papers not cited in the AR6 (including but not limited to Mann et al. (2010), Chen et al. (2022), Everett et al. (2022), Kelesidis et al. (2022), Sedlacek et al. (2022), Ramachandran et al. (2023). Major causes of these differences are due to complexities in terms of both microphysical representations of BC on a per-particle basis, as well as emissions and column loading of BC on an atmospheric basis, meaning that the discussion continues to have profound implications for global warming and regional climate change, as well as on the remote sensing retrieval community.*"

We thank the reviewer for this insightful correction. The term "secondary aerosol vapors" was originally intended to describe the gaseous species that condense onto black carbon (BC) particles during the atmospheric aging process. However, we agree that "precursors" is the more precise and standard term used in atmospheric science to describe the gaseous chemicals (such as $SO_2$, $NO_x$, and VOCs) that undergo chemical transformation and subsequent condensation. We also need to consider that our computed shell is done optically, and hence will also include water in the aerosol phase, and therefore have done some re-writing in the text to more appropriately capture the processed underlying the observed shell size and mixing state properties.

We have updated the sentence in Line 42 to reflect this professional terminology and ensure clarity:

"*... other primary aerosols, secondary aerosol precursors, and atmospheric water vapor, forming a 'core-shell' type of structure that is thermodynamically stable.*"

L58-59: this is not the case for studies such as Li et al., 2019.

We thank the reviewer for highlighting the advanced work of Li et al. (2019). We agree that component-based retrieval frameworks (such as GRASP) have significantly improved upon traditional methods by dynamically retrieving aerosol components. Our study provides a complementary approach by leveraging the unique information content of MISR observations. The multi-angle heritage of MISR (9 viewing angles) provides a distinct possibility to better constrain the microphysical state of BC. By using MISR's multi-band observations, we can explore the variability of BC's size distribution and mixing state in a way that provides additional insights into its global distribution. In specific, we are able to produce a core size at 10nm probabilistic resolution, a shell size at 10nm probabilistic resolution, hence a variable mixing state across the size distribution, and a wavelength variable SSA which are not homogenous and vary in ways not constrained by using traditional angstrom exponent approaches. However, this approach requires an initial set of solutions from a component-based retrieval framework such as GRASP. Henceforth, improvements in GRASP can be used

to further refine the approach used herein, allowing us to produce even better products in the future.

We have updated the sentence in Lines 68-75 to acknowledge these advanced studies while clarifying the specific possibility provided by the MISR-based approach:

"*Traditional approaches have frequently used retrievals of aerosol optical depth from satellite observations to derive BC column mass concentrations using fixed and/or simplified microphysical properties of BC. While sophisticated component-based retrieval frameworks (Li et al., 2019) have made advancements in this field including better use of polarization and improved representation of particle size, the use of multi-angle and multi-band observations from MISR provide a unique possibility to further constrain the microphysical variability of BC, such as high resolution particle size and mixing state variability, and non-homogenous computation of SSA, allowing both more realistic per-particle microphysical and total atmospheric column constraints on BC on a global scale.*"

L64: -> have been proven

We thank the reviewer for this grammatical correction. We have updated the sentence in Line 79 to ensure grammatical accuracy and clarity:

"*Numerical simulations have been proven invaluable in investigating the sources, transport, and deposition processes of BC column concentrations.*"

L75: what do you mean daily preprocessing?

We thank the reviewer for raising this question and fully agree that the original phrasing "daily preprocessing" was unclear and could be misinterpreted. Our intention was not to imply a specialized preprocessing algorithm, but rather to indicate that MISR observations are read and processed on a day-by-day basis to extract the necessary input parameters for the inversion. To avoid ambiguity, we have updated the operational description to more accurately reflect the steps of the method and remove the misleading implication of a distinct "preprocessing" stage. We appreciate the

reviewer's attention to this detail, as the clarification improves the accuracy and readability of the manuscript.

We have revised the sentence at Line 89 as follows:

"*Specifically, the method involves leveraging data from four spectral bands of the MISR satellite by reading and processing them on a daily basis to obtain critical parameters, including Absorption Aerosol Optical Depth (AAOD) and Single Scattering Albedo (SSA).*"

L80: validate with MERRA-2 reanalysis seems to be the other way around.

We thank the reviewer for this insightful comment. We agree that "validate" is not the appropriate term for comparisons with the MERRA-2 reanalysis, as MERRA-2 is a model-based product that carries its own set of uncertainties. One of MERRA-2's largest issues is that it does not explicitly assimilate AAOD or SSA observations, nor does it incorporate spectral aerosol column information into its data assimilation system (Randles et al., 2017; Buchard et al., 2017), meaning that the comparisons provided serve to highlight how this additional information leads to differences.

Regarding the discrepancies between our results and MERRA-2, we believe these differences highlight the value of our approach. While MERRA-2 relies heavily on a priori emission inventories and fixed transport modeling which may not be consistent in space and time with real emissions on the ground, our dataset is constrained by MISR observations. These discrepancies demonstrate the need for an independent, observation-based BC product, especially in regions where a priori model information may be inconsistent or out of date with actual emissions conditions.

We have revised the sentence in Lines 95-98 to use more accurate terminology, focusing on inter-comparison and consistency rather than formal validation:

"*Furthermore, comparisons with AERONET optical properties and inter-comparisons with the MERRA-2 reanalysis serve to evaluate and demonstrate the method's consistency, realism over areas with limited, misplaced, or out of date a priori*

*emissions data, and applicability to offering a new consistent product across global scales at daily resolution over decadal temporal scales.*"

Section 2.1: the first paragraph misses important references about MISR. The same for entire Section 2.1, it's difficult for me to imagine that you used MISR data without citing any references.

We thank the reviewer for emphasizing the need to properly reference the MISR instrument and its data products in Section 2.1. We fully agree that the earlier version did not provide adequate citations to support the description of MISR observations, which could give the impression that the dataset was introduced without reference to the extensive literature documenting its design, retrieval algorithms, and validation. Please see detailed response given above to the nearly same question.

L135-150: I find this section meaningless, as it is merely a repetition of the classical Mie theory. Reiterating this part does not seem to contribute to the study, and moreover,

We appreciate the reviewer's careful assessment of this subsection and fully agree that the previous presentation of classical Mie theory in Lines 135-150 may be considered of lesser importance. In response, we have removed the entire block of generic theoretical derivations from this subsection. Rather than repeating textbook content, the revised section now focuses exclusively on the components of the Mie calculation that are directly relevant to our inversion framework, including the specific optical parameters used, the assumed core-shell configuration, the aerosol core and shell size analysis approach, the wavelength-dependent refractive indices, and the way these elements interface with MISR's multi-angle SSA/AAOD observations across different wavebands. We have also added proper references to the established Mie-scattering literature and to studies that employ similar parameterizations in aerosol retrievals. The revised subsection therefore provides a clear, concise, and logically motivated methodological description that is directly tied to our study's inversion approach, fully addressing the reviewer's concern that the previous version lacked purpose, relevance, and citation support.

L153: retrievals are not observations

We thank the reviewer for noting this terminology issue. We agree that "retrievals" is the correct term in this context, as MISR aerosol products are derived quantities rather than direct observations. Accordingly, we have revised the wording at Line 154 to replace "observations" with "retrievals", ensuring that the description is conceptually accurate at that specific location.

*"..., this study utilizes multispectral remote sensing retrievals from MISR."*

L160: I find out that there's no discussions about MISR AAOD accuracy, the authors can easily validate MISR AAOD or SSA with AERONET.

We thank the reviewer for this constructive point. We fully agree that establishing the accuracy of the input MISR products (AAOD and SSA) is a necessary foundation for the subsequent BC retrieval.

As the reviewer noted, validating these products with AERONET is a direct and effective way to ensure this reliability. While the MISR aerosol products have been extensively validated on a global scale by the MISR science team (Kahn et al., 2010; Garay et al., 2020), we recognize the importance of demonstrating their performance specifically for the data used in our framework.

Therefore, in the revised manuscript, we have added a dedicated discussion regarding the established accuracy of MISR products. We have found that AERONET uncertainties for SSA are 0.03 (Dubovik et al., 2000) and for AOD are +0.06 to -0.02 (Giles et al., 2019), and that MISR uncertainty for AOD is +0.05 or 0.2 × AERONET (Garay et al., 2020; Kahn et al., 2010). We believe that the MISR SSA uncertainty must be at least as large as AERONET SSA uncertainty, which should be about 20%-40% under conditions where the SSA is in the range of 0.8 to 0.95. Furthermore, we have highlighted the comparison between MISR-derived AAOD and AERONET observations as a specific quality control step, and confirm that the differences in terms of RMSE and MAE are found to be within a range of ~64% (assuming an SSA of 0.92), which is smaller than or similar to the combined MISR uncertainties of 20% × AOD and another 20%-40% due to SSA. This confirms that our input parameters align well with ground-based benchmarks, providing a solid basis for the BC results.

We have revised the text in Lines194-204 as follows:

"*The accuracy of MISR products has been established through extensive global validation against AERONET (Kahn et al., 2010; Garay et al., 2020), providing a physical foundation for our retrieval. In terms of specific error budgets, AERONET reports uncertainties of 0.03 for SSA and +0.06 to -0.02 for AOD. In comparison, the MISR AOD uncertainty is established at +0.05 or 0.2 × AERONET AOD. Given these constraints, the MISR SSA uncertainty is estimated to be at least as large as the AERONET SSA uncertainty, translating to approximately 20%–40% under typical conditions where this process is successful, the SSA ranges from 0.80 to 0.95. To further ensure the robustness of our inputs, we performed a consistency check by comparing MISR-derived AAOD with AERONET observations (Figure B1 and Table C4). The resulting differences have Root Mean Square Error (RMSE) and Mean Absolute Error (MAE) statistics both falling within approximately ~64% (assuming an SSA of 0.92), which is the combined uncertainty range (propagated using 20% AOD and 20%–40% SSA uncertainties). This alignment with ground-based benchmarks confirms that the input parameters provide a solid basis for reliable BC estimation.*"

L192: ±0.03 doesn't mean the uncertainty of MISR SSA products used in your calculation, you'd better to evaluate or refer to specific MISR studies.

We thank the reviewer for the constructive suggestion regarding the SSA uncertainty. The decision to use ±0.03 as the uncertainty for MISR SSA was based on the physical limits of current observation systems and established literature. AERONET is widely recognized as the primary benchmark for SSA, with an established uncertainty of approximately 0.03. From a physical retrieval perspective, satellite-based estimates (like MISR) are subject to additional uncertainties from surface reflectance and atmospheric path variations, making it unlikely for them to achieve higher precision than synchronized ground-based measurements.

Assuming that satellite uncertainty is at least as high as the best available ground-based benchmark (0.03) represents a conservative and logical scientific approach. Furthermore, previous evaluations of comparable satellite products (e.g., TROPOMI and OMI) for carbonaceous aerosols indicate that SSA retrieval errors typically range

between 0.03 and 0.05. Following the reviewer's suggestion, we have now incorporated references to specific MISR studies and expert assessments (Kahn et al., 2010; Kalashnikova et al., 2013) to provide a clearer physical context for the uncertainty levels used in our calculations. We observe that comparisons between our product AAOD and AERONET AAOD over the Amazon have RMSE and MAE errors which are always within ~64%, meaning they are consistent with an SSA uncertainty of 0.03.

We have revised the sentence at Lines 240-244 as follows:

"*Observed values from each band defined a possible solution space by considering the ±0.03 uncertainty of SSA observations (Torres et al., 2020; Tiwari et al., 2025), to account for inherent observational uncertainties and instrument noise. This uncertainty level represents a conservative physical limit aligned with the AERONET ground-based benchmark of ~0.03, although we acknowledge that in fact it may be slightly larger. Under this framework, only grid points that met the range requirements across all four bands in tandem are retained.*"

L198: I don't find detailed description of about your method, and another key is how you assume BC refractive index?

We thank the reviewer for raising this important point. We agree that the earlier version of the manuscript did not provide a sufficiently detailed description of the inversion procedure, nor did it adequately document the assumptions used for the BC refractive index.

In the revised manuscript, we have substantially expanded the methodological exposition around Line 232 to clearly explain each component of the retrieval framework. This includes an explicit description of the specific optical calculations used, the physical assumptions applied to BC cores and their coatings, and the manner in which MISR SSA and AAOD are incorporated as constraints. Most importantly, we now provide a clear statement of the wavelength-dependent refractive indices adopted for BC, together with citations to Schuster et al. (2005) and other foundational aerosol-optics studies that justify these parameter choices. The revised text also clarifies the

assumed BC and coating densities and the core–shell representation applied in the optical calculations.

These additions address the reviewer's concern by making the methodological assumptions fully transparent and scientifically grounded, ensuring that readers can understand, evaluate, and reproduce the inversion method.

Section 3.3: before in-depth discussion about the global BC results, I would see more about the method.

We thank the reviewer for this valuable comment. We agree that the earlier version of the manuscript introduced global BC results before providing a sufficiently detailed methodological foundation, which made it difficult for readers to properly assess the robustness of the retrievals. In response, we have thoroughly reorganized and expanded the Methods section to ensure that all methodological content is presented in a clear, comprehensive, and logically structured manner before any discussion of results. Specifically, all details related to the optical calculations, core-shell particle representation, parameter assumptions, and the use of MISR SSA and AAOD constraints have been moved from the original Section 3 into an expanded Section 2. This section now includes explicit definitions of all variables, full descriptions of the refractive index and density assumptions for BC and coating materials. We also revised the relevant text to clarify the physical basis of the inversion and to ensure full transparency regarding the assumptions and computational steps.

With these revisions, Section 3 now focuses exclusively on dataset-oriented results, including the spatial patterns, temporal behavior, and derived microphysical characteristics, without mixing in methodological explanations. This restructuring presents the material in a more coherent order and aligns the manuscript in which methods precede results and are described with sufficient detail to ensure reproducibility. Collectively, these changes provide a stronger methodological foundation for the interpretation of global BC fields and fully address the reviewer's concern.

**References**

Bond, T. C. and Bergstrom, R. W.: Light Absorption by Carbonaceous Particles: An Investigative Review, Aerosol Science and Technology, 40, 27–67, https://doi.org/10.1080/02786820500421521, 2006.

[revised manuscript text omitted]